# Catch the star! Spatial information activates the manual motor system

**A. Miklashevsky**(iD)*

Potsdam Embodied Cognition Group, Cognitive Sciences, Faculty of Human Sciences, University of Potsdam, Potsdam, Germany

* armanster31@gmail.com

## Abstract

Previous research demonstrated a close bidirectional relationship between spatial attention and the manual motor system. However, it is unclear whether an explicit hand movement is necessary for this relationship to appear. A novel method with high temporal resolution–bimanual grip force registration–sheds light on this issue. Participants held two grip force sensors while being presented with lateralized stimuli (exogenous attentional shifts, Experiment 1), left- or right-pointing central arrows (endogenous attentional shifts, Experiment 2), or the words "left" or "right" (endogenous attentional shifts, Experiment 3). There was an early interaction between the presentation side or arrow direction and grip force: lateralized objects and central arrows led to a larger increase of the ipsilateral force and a smaller increase of the contralateral force. Surprisingly, words led to the opposite pattern: larger force increase in the contralateral hand and smaller force increase in the ipsilateral hand. The effect was stronger and appeared earlier for lateralized objects (60 ms after stimulus presentation) than for arrows (100 ms) or words (250 ms). Thus, processing visuospatial information automatically activates the manual motor system, but the timing and direction of this effect vary depending on the type of stimulus.

## Introduction

The manual motor system and orienting of spatial attention are closely related. Eye-hand coordination enables efficient reaching, grasping, and object manipulation. This coordination requires analyzing visual information about the environment surrounding hands and simultaneous control of ongoing hand movements [see for review 1].

Attention can prioritize spatial locations already before the execution of a hand movement. This effect was shown in a series of experiments where participants pointed at different locations on a screen and then reported a briefly presented target symbol [2, 3,see also 4]. Notably, the target symbol was presented either in a position later pointed at or in a different position. Participants were significantly more accurate at identifying the target symbol when it was presented in a to-be-pointed-at location. Eye-hand coordination is a complex bidirectional process and not merely serial delivery of motor commands to the execution level after complete analysis of visual information. Multiple target coordinates are averaged and influence the

**Data Availability Statement:** The datasets generated for this study can be found in the Open Science Framework (DOI: 10.17605/OSF.IO/H6P7M).

**Funding:** I acknowledge the support of the Deutsche Forschungsgemeinschaft (DFG) and

Open Access Publishing Fund of the University of
Potsdam. The funders had no role in study design,
data collection and analysis, decision to publish, or
preparation of the manuscript.

**Competing interests:** The author has declared that
no competing interests exist.

ongoing movement, constantly updating the motor program [5, 6, see 7, for a theoretical account].

In line with this research is the so-called near-hand effect [8]: attention is more engaged when objects appear close to hands. Participants more efficiently inhibit distractors presented farther away from hands [9] and identify new objects in the near-hand space faster [10, see 11, for a review]. Hand movements lead to dynamic changes of the near-hand space, which results in faster detection of targets closer to hands at every moment [12]. Also, the task and type of stimuli matters: having one's hand close to words or arithmetical expressions is disadvantageous for semantic processing of those symbolic stimuli [13, 14].

Thus, previous research convincingly demonstrated a bidirectional relationship between spatial attention and hand movement across various paradigms. But how automatic is this link between the manual motor system and spatial processing? Is hand movement, whether ongoing or potential, a necessary component for this relationship to appear? In a previous study, my colleagues and I presented participants with large vs. small numerical stimuli [15] while monitoring participants' spontaneous hand motor activity using two grip force sensors [16–20, for review and methodological details, see 21]. No manual response or explicit hand movement was required. Numerical cognition research shows that small numbers are associated with the left peripersonal space, and large numbers are associated with the right peripersonal space [Spatial Numerical Association of Response Codes, or SNARC effect, see 22]. This effect has been demonstrated with button press responses, finger movements [23], eye movements [24], foot responses [25], and even full-body movements [see for reviews 26, 27, see for meta-analysis 28]. Despite this overwhelming evidence for the presence of spatial-numerical associations, we found no SNARC effect in grip force [15], either because no attentional shifts appeared without explicit motor responses to numeric stimuli [cf. 29] or because the manual motor system is not activated automatically by spatial information. The latter hypothesis was tested explicitly in the present study.

The goal of the present study was to investigate the direct effects of attentional shifts on the manual motor system in the absence of any motor action. A novel method, bimanual grip force recording, allows monitoring motor activity in hands with millisecond resolution while participants process visual stimuli. Grip force sensors measure spontaneous motor activity during action observation [30] and semantic processing of motor-related language [16–20]. Both hands exhibit comparable activity in response to motor-related linguistic stimuli [31].

In the present study, I recorded grip force bimanually while manipulating participants' visual attention. Specifically, exogenous and endogenous attentional shifts [see 32, 33] were induced using different types of stimuli. Lateralized stimuli cause exogenous attentional shifts by summoning attention as they are physically present at a particular location. In the present study, exogenous attentional shifts were induced by presenting participants with left- or right-localized stars (Experiment 1). In contrast, centrally presented symbolic stimuli direct attention by their meaning and thus lead to endogenous attentional shifts. In the present study, endogenous attentional shifts were induced by presenting participants with left- or right-pointing arrows (Experiment 2) or words "left" and "right" (Experiment 3). Words produce attentional effects different than those resulting from arrows. While some authors assumed that both arrows and linguistic cues cause similar endogenous attentional shifts [e.g., 33], more recent studies demonstrated essential differences between these types of cues [34]. A series of studies showed that words with a typical localization in vertical space (e.g., shoe, grass, sun, or cloud) could shift attention in the corresponding direction [35–37]. This effect extends even to abstract concepts, such as god or devil [38]. Simple linguistic cues (e.g., the word *up*) can also modulate trajectories of vertical saccades [39].

Additionally, bimanual grip force recording has a high temporal resolution (1000 Hz), making it possible to reveal the precise timing of manual motor activity accompanying attentional shifts. I expected the effect of interest to emerge later for symbolic stimuli (arrows and words) than for physical objects (stars), since symbols require one additional processing step–extraction of meaning–for causing attentional shifts.

## General method

Forty-two psychology and linguistics students (14 males, 28 females) of the University of Potsdam participated in the study for course credit. An entire testing session consisted of four experiments: lateralized star presentation (Experiment 1), lateralized sound presentation (not reported here), central arrow presentation (Experiment 2), and word presentation (Experiment 3). The force pattern generated by sound presentation was strikingly different from those resulting from other types of stimuli: sounds lead to a clear initial dip in force followed by one short and one very long peak, probably correlating in magnitude with sound intensity. It seems that qualitatively different processes are reflected in grip force in this case.

All experiments were conducted in a pseudorandom order to exclude possible systematic sequence or fatigue effects. The exact order of experiments is specified for every participant in supplementary materials (see data availability statement). Participants were allowed to take breaks between experiments, walk and drink water. The whole testing session lasted between 60 and 90 minutes.

After the experiments, participants completed questionnaires including demographic data (gender, age, native language, and foreign languages they speak) and physiological data (seeing problems, hearing problems, motor diseases, and whether they take medications that can influence motor control). Additionally, participants filled in the Edinburg Handedness Inventory [EHI, 40], where original instructions were replaced with a more intuitive Likert scale as suggested elsewhere [41]. Resulting EHI scores range from +100 (exclusively right-handed) through 0 (ambidextrous) to -100 (exclusively left-handed). All participants signed an informed consent form before the study.

The study was approved by the Ethics Committee at the University of Potsdam (www.uni-potsdam.de/de/senat/kommissionen-des-senats/ek; study number 15/2019).

One participant was excluded due to a self-reported motor problem (light essential tremor) clearly reflected in his force data. Another participant reported left leg paralysis, but her data were not qualitatively different from the rest of the sample and thus remained in the final dataset. Only German native speakers participated in the word presentation study (Experiment 3). All but one participant reported having normal or corrected-to-normal vision.

## Equipment and data acquisition

The method followed closely the one recommended by Nazir et al. [21] for single-sensor recording. Both sensors were stand-alone load cells manufactured by ATI Industrial Automation, USA (www.ati-ia.com/Products/ft/sensors.aspx). They resembled large metal coins with 40 mm diameter and 14 mm height, and each weighed 57 g. Each sensor was covered from both contact sides with a 3 mm plastic cover of the same diameter as the sensor itself (40 mm), resulting in a total thickness of 20 mm and a total weight of 65 g per sensor (see Fig 1). These sensors record force dynamics with millisecond resolution along three orthogonal axes, but only Fz force along the vertical axis through the sensors was analyzed and reported here. Two PCs were used: one for running the experiment under OpenSesame software [42] and another for force data acquisition under Expyriment software [43]. The first PC sent a trigger at the beginning of each trial to later identify a corresponding time point in the force data file.

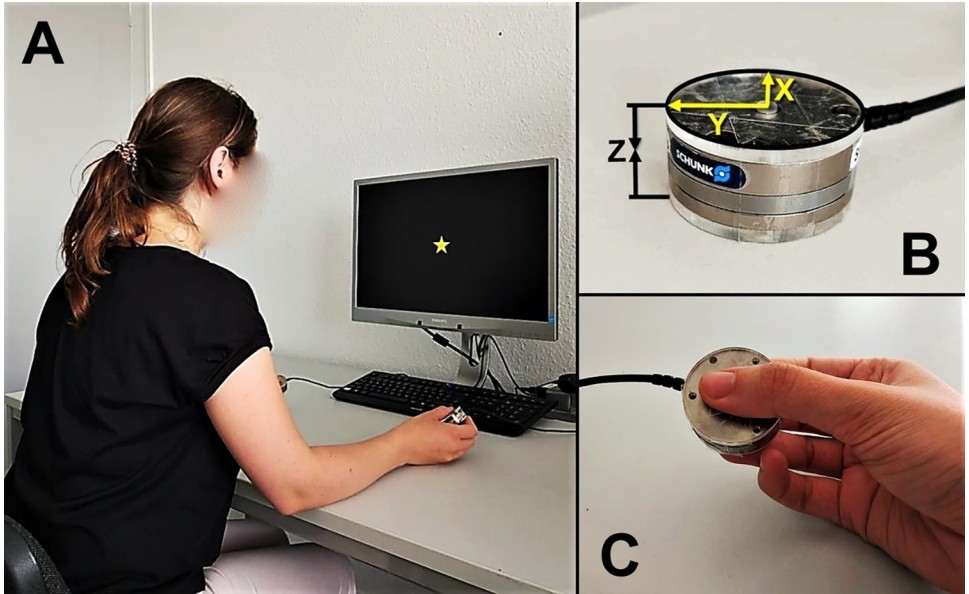

**Fig 1. Experimental setup and equipment.** Panel A. Bimanual force recording setup. Panel B. Grip force sensor (X–longitudinal, Y–radial, Z–compression forces). C. The way participants held the sensors (around 45° relative to the table surface, not strictly controlled). Adapted image from Miklashevsky et al. [15, Fig 2].

Monitor Philips Brilliance LCD 220P4LPY 22" with a screen resolution 1680x1050 was used for stimuli presentation.

Participants sat at a desk and held one sensor in each hand at an angle of around 45° relative to the table surface with the thumb on one side and the index and middle fingers on the other side. Participants' elbows rested on the table while their hands held the sensors, thus preventing sensor slippage (see Fig 1). The distance between sensors varied from 30 to 50 cm and was not strictly controlled, but both were equidistant from each participant's mid-sagittal plane. The distance from participants' eyes to the screen was around 60 cm.

Before data collection, participants practiced applying a holding force in a range between 1.5 N and 3 N with each hand. The sensors were represented on the screen as two circles that changed their color from green ("too weak") to red ("too strong") with the pre-defined force range indicated by the grey color. As soon as participants managed to turn both circles into grey, they received an instruction to keep the force at this level during the whole testing session. Data collection started automatically after participants held the sensor with the required force for three seconds without crossing these thresholds. This calibration procedure was repeated after each break and at the beginning of each experiment. Most participants successfully learned to perform the calibration within 15–30 seconds. There were, however, a few participants who required up to two minutes during the first calibration.

There was no cover story used for the participants.

## Experiment 1: Star presentation

Participants were presented with either central or lateralized (left vs. right) visual stimuli in this experiment. This ensured exogenous control over attention [33], while bimanual force recording allowed investigating the dynamic involvement of the motor system in the processing of spatial information.

## Participants

Forty-one psychology and linguistics students (13 males, 28 females; mean age = 24 years) participated in the experiment. Their mean EHI score was 66 (80% had EHI score > 50; 10% had EHI score between +50 and -50; 10% had EHI score < -50). All but one participant reported normal or corrected-to-normal vision. No participant took medications affecting motor control.

## Stimuli and design

Red and yellow stars were used as stimuli in catch (go) and critical (no-go) trials accordingly. The background was kept black. Stimuli can be found in the supplementary data (see data availability statement). Stars appeared in the middle of the screen, on the left or the right side. Grip force was recorded bimanually. This results in a 2 (Hand: left / right) X 3 (Position of the star: left / central / right) within-participant design.

## Task and procedure

After the calibration procedure described above, the experiment started. Each trial consisted of a fixation dot (200 ms), followed by a stimulus (until response, but no longer than 2000 ms). The stimuli were stars around 4 cm in diameter (3.82 degrees of visual angle calculated by using the formula 57.3*w/d; w–width of the object; d–distance to the object), which appeared with equal probability (33%) in one of three positions: at the center of the screen, or 19.5 cm left or right from the center (18.62 degrees of visual angle). 75% of the stars were yellow, and red stars appeared in 25% of all trials. The task was to say "yes" when a red star appeared, regardless of stimulus location. Participants were asked not to rotate their heads when stars appeared laterally, but eye movements were allowed.

Additionally, participants were instructed not to cross their legs during the experiment. Critical trials for analysis were no-go trials (yellow stars). This means that overt motor or verbal responses do not contaminate grip force recordings. Such responses typically generate large artifacts in these recordings (e.g., see panel A at Fig 2). The experiment consisted of 360 trials with a break in the middle and lasted around 15 minutes. It was preceded by a short practice (12 trials).

## Data preprocessing and analysis

The preprocessing of grip force data closely followed the recommendations of Nazir et al. [21, Experiment 2]. Data were filtered at 15 Hz before analysis with a fourth-order, zero-phase, low-pass Butterworth filter. Single epochs were extracted from the vertical Fz signal, starting 200 ms before and ending 1000 ms after stimulus onset. The global variation in force across the experiment was corrected by (1) averaging force within a 20 ms interval before stimulus onset in each epoch and (2) subtracting this average force from the entire epoch. As a result, grip force always crosses the zero point at the start of each trial, and negative force values reflect a vertical grip force less than that at the moment of stimulus presentation, not the absence of force. Under global force variation, I understand here stimulus-unrelated changes in force over longer periods of time, e.g., if participants press sensors stronger due to higher arousal at the beginning of the experiment or slightly release the sensors in the second half of the session due to fatigue, etc.

Maximum and minimum thresholds were applied (±500 mN) to remove movement artifacts and identify participants with unacceptably large force variability. The proportion of trials where force exceeded one of the thresholds varied across participants from 0% to 19%

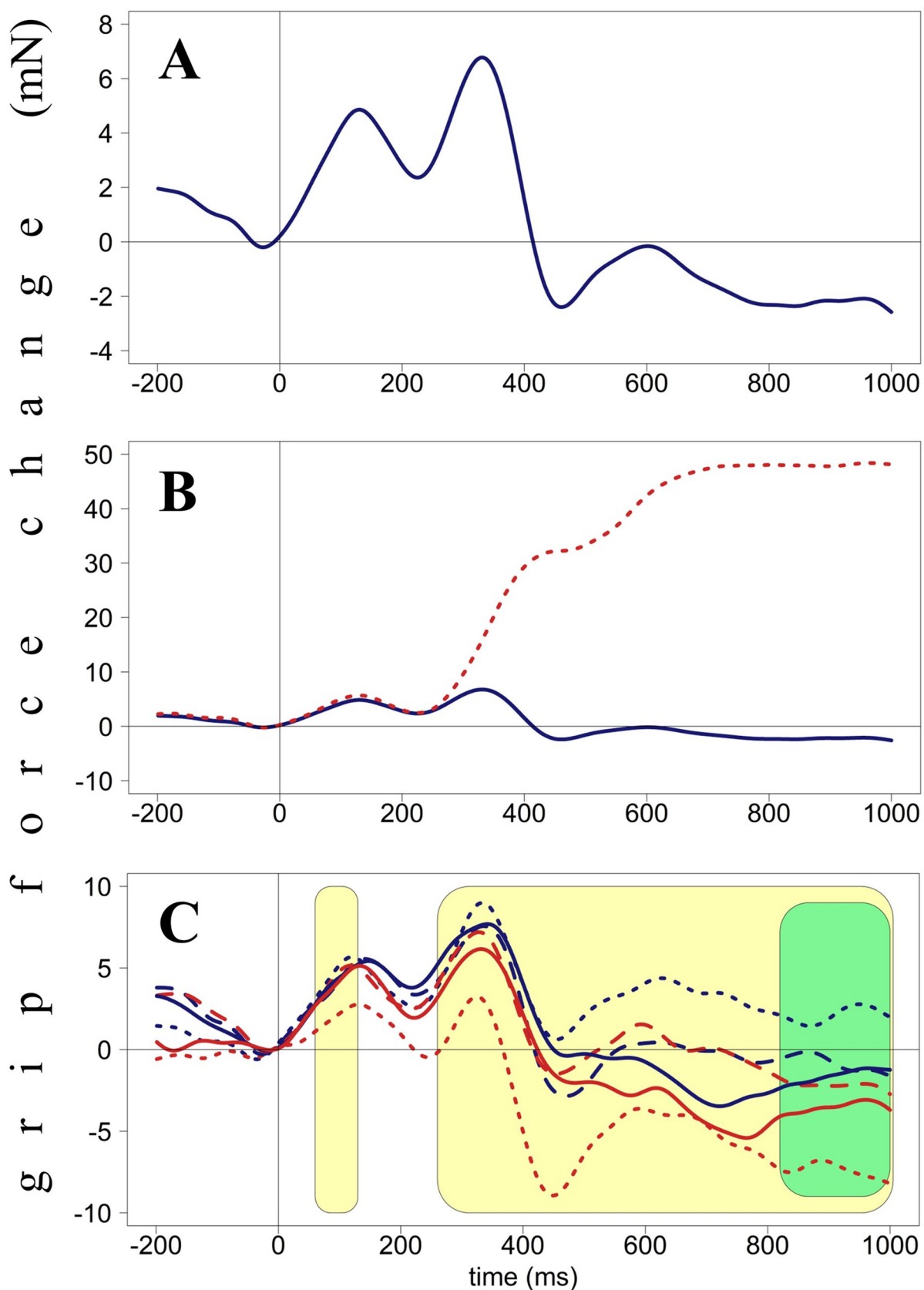

**Fig 2. Grip force changes (in milli-Newton) plotted against time from stimulus onset (in milliseconds), Experiment 1.** Panel A.
Averaged force profiles across all accepted no-go trials of all participants. Panel B. Force profiles in go (dotted red line) and no-go (solid blue
line) trials. These forces diverge around 250 ms after stimulus presentation. Panel C. Force profiles averaged by condition (Hand X Position).
Light-yellow areas (60–130 ms and 260–1000 ms) indicate interactions between Hand and Position, green area (820–1000 ms) indicates the
main effect of Hand. Red lines represent right-hand forces, blue lines–left-hand forces. Dotted lines represent the star-left condition, solid
lines–star-in-the-center condition, dashed lines–star-right condition.

(mean = 2%). Those trials were discarded, but no participant was excluded due to this crite-
rion. Accuracy varied from 98% to 100% (mean = 100%); error trials were excluded from fur-
ther analysis.

Overall grip force patterns are presented in Fig 2 to give readers an overview of this rela-
tively unfamiliar data type. The blue line in Fig 2A represents averaged force of both hands for
all accepted no-go trials. As suggested before [15], these changes will be referred to as H (high
force, peaks), with a number representing a time point. For example, H130 means a peak with
its highest point at around 130 ms after stimulus onset. As one can see, independently of a par-
ticular condition, grip force produces three peaks (H130, H330, and H600). The second peak
(H330) is the tallest. Fig 2B represents averaged grip force changes in go (dotted red line) and
no-go (solid blue line) trials. These two lines start diverging at 250 ms after stimulus onset with
force in go trials reaching its highest point (almost 50 mN) at around 750 ms and remaining at
this level till the end of the epoch (1000 ms). Fig 2C represents force averaged by condition
(Hand and Position): red lines represent right-hand forces, blue lines represent left-hand
forces; dotted lines represent the star-left condition, solid lines the star-in-the-center, dashed
lines the star-right condition.

The method used by Miklashevsky et al. [15] was applied to identify time windows of inter-
est: forces were aggregated by Hand (left / right) and Position (right/central/left) within partic-
ipants. With Hand (left / right) and Position (left / center / right) as within-variables and
interaction between them, these data were submitted to a cluster permutation analysis [44,
package "permuco", see 45]. Five thousand permutations were performed, and TFCE (Thresh-
old-Free Cluster Enhancement) correction for multiple comparisons was used. The analysis
revealed four time windows with significant or close to significance effects: main effect of
Hand (820–1000 ms after stimulus onset), main effect of Position (680–760 ms), and two time
windows with interactions between Hand and Position (60–130 ms and 260–1000 ms after
stimulus onset; see Fig 3).

Cluster permutation analysis is a bootstrapping method for continuous signal. In this analy-
sis, conditions are randomly assigned to epochs, which results in a random data structure.
According to newly assigned labels, a t-statistics is calculated. The mass of the clusters exceed-
ing a significance threshold is stored. The procedure is repeated multiple times, with the result-
ing distribution of random cluster masses that can be found in the dataset. The actual cluster
mass is then compared with bootstrapped cluster masses, and the likelihood of the observed
result is calculated. I suggest using this method in case of force registration for exploratory pur-
poses to identify potential time windows of interest. Linear mixed-effects models are used in
the present study for confirmatory analyses.

**820–1000 ms, Model 1.1.** The data (averaged by Hand and Position for each participant)
were then submitted to a linear mixed model analysis using the lme4 package [46] in R. The
categorical predictor Hand was sum-coded [left/right, sum-coded contrast -0.5 and 0.5, see
47]. Position was re-coded in a continuous manner (left: -1; center: 0; right: 1). Interaction
between Hand and Position was included. Participants were included as random factors. I per-
formed a backward elimination using the drop1 function to identify the best-fit model; effects
and interactions that did not improve model fit (p > .1) were successively eliminated. Only

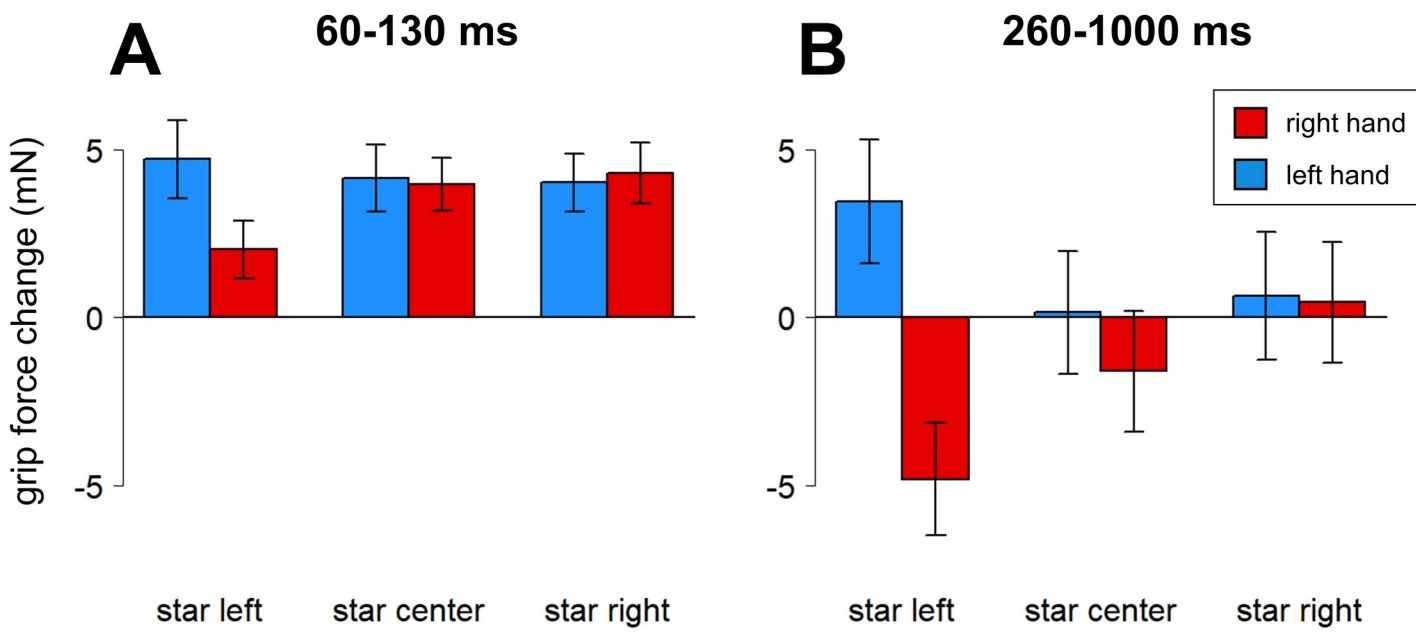

**Fig 3. Grip force changes (in milli-Newton) averaged by hand and position, Experiment 1.** Panel A. Average forces in time widow 60–130 ms. Panel B. Average forces in time widow 260–1000 ms. Whiskers represent standard errors.

significant effects are reported unless the effect of interest in a given analysis was non-signifi-cant–in such cases, it is also reported.

The main effect of Hand was significant: grip force in the left hand was more prominent than in the right hand (b = -4.261, p = .001). The interaction between Hand and Position was also significant (b = 4.071, p = .009). Still, since this time window is a part of a larger one (260–1000 ms) suggested by the cluster permutation analysis for this interaction, it was not exam-ined further. Marginal r-squared [variance explained by fixed effects, see 48] was .041, and conditional r-squared (variance explained by the whole model, i.e., fixed and random effects together) was .451. See Table 1 for further details.

**680–760 ms, Model 1.2.** The same model and approach were used as before but with averaged force in the time window 680–760 ms as a dependent variable. The effect of Hand was significant (larger force in the left hand: b = -3.331, p = .017), as well as interaction between Hand and Position (b = 4.291, p = .012). Note that this time window is a part of a larger one (260–1000 ms), where this interaction was investigated in detail (see below). The main effect of Position was not significant (b = 0.382, p = .655). Marginal r-squared was .026, and conditional r-squared was .473. See Table 2 for further details.

**Table 1. Model 1.1.** Main effect of hand on grip force in the time window 820–1000 ms (Experiment 1).

| Random effects: | Name | Variance | SD | |
|---|---|---|---|---|
| Participants | Intercept | 74.890 | 8.654 | |
| Residual | | 100.180 | 10.009 | |
| **Fixed effects:** | **b** | **SE** | **t-value** | **p-value** |
| Intercept | -2.230 | 1.495 | -1.492 | .136 |
| **Hand** | **-4.261** | **1.276** | **-3.339** | **.001** |
| **Hand*Position** | **4.071** | **1.563** | **2.604** | **.009** |

**Table 2. Model 1.2.** Main effect of Position on grip force in the time window 680–760 ms (Experiment 1).

| Random effects: | Name | Variance | SD | |
|---|---|---|---|---|
| Participants | Intercept | 102.000 | 10.100 | |
| Residual | | 120.100 | 10.960 | |
| **Fixed effects:** | **b** | **SE** | **t-value** | **p-value** |
| Intercept | -1.692 | 1.725 | -0.981 | .327 |
| **Hand** | **-3.331** | **1.398** | **-2.384** | **.017** |
| **Position** | **0.382** | **0.856** | **0.446** | **.655** |
| **Hand*Position** | **4.291** | **1.712** | **2.507** | **.012** |

**60–130 ms, Model 1.3.** The data were restructured, and the mean-centered force of the opposite hand in the same time window was included as a predictor. The force of the opposite hand was used as a predictor in order to account for the correlation of forces due to automatic coordination between hands [49]. I expect lateralized stimuli to lead to increased force on the ipsilateral side. Still, at the same time, it is clear from force patterns that the force of both hands changes simultaneously, i.e., when the force of one hand increases, so does the other (see Figs 2C, 6C, and 9C). It implies that this basic physiological mechanism might mask lateralized effects of interest. That is why I suggest using the contralateral force as a covariate and estimating the effect of interest beyond the variance explained by the contralateral force.

As before, continuously coded Position (-1 = left, 0 = center, +1 = right) was included as fixed effect, participants were included as random intercepts. Function drop1 was used to identify and successively eliminate non-significant terms. The effect of Position on the left-hand force (after accounting for the contralateral force) was close to significance with higher force when stars were presented on the left side (b = -0.681, p = .052; marginal r-squared = .073, conditional r-squared = .726; see Table 3 for details; see also Fig 4).

**Model 1.4.** A similar analysis was run with the right-hand force as a dependent variable and Position and mean-centered left force as fixed predictors. This time a strong effect of Position was found with larger right force when stimuli were presented on the right side (b = 1.316, p = .005; marginal r-squared = .371, conditional r-squared = .408; see Table 4 for details; see also Fig 4).

260–1000 ms, Models 1.5 and 1.6

The same approach was used as for the previous time window (60–130 ms). Each force was tested separately, with the contralateral force and Position as predictors. This time the effect of Position was significant in both hands: the left force increased when stimuli were presented on the left (b = -1.989, p = .011; marginal r-squared = .053, conditional r-squared = .665; see Table 5) and the right force increased when the stimuli were presented on the right (b = 2.949, p < .001; marginal r-squared = .092, conditional r-squared = .621; see Tables 5 and 6; see also Fig 5).

**Table 3. Model 1.3.** Effect of Position on the left grip force (after controlling for the contralateral force) in the time window 60–130 ms (Experiment 1).

| Random effects: | Name | Variance | SD | |
|---|---|---|---|---|
| Participants | Intercept | 22.434 | 4.736 | |
| Residual | | 9.435 | 3.072 | |
| **Fixed effects:** | **b** | **SE** | **t-value** | **p-value** |
| Intercept | 4.305 | 0.790 | 5.451 | < .001 |
| **Contralateral hand (right)** | **0.288** | **0.074** | **3.892** | **< .001** |
| **Position** | **-0.681** | **0.350** | **-1.947** | **.052** |

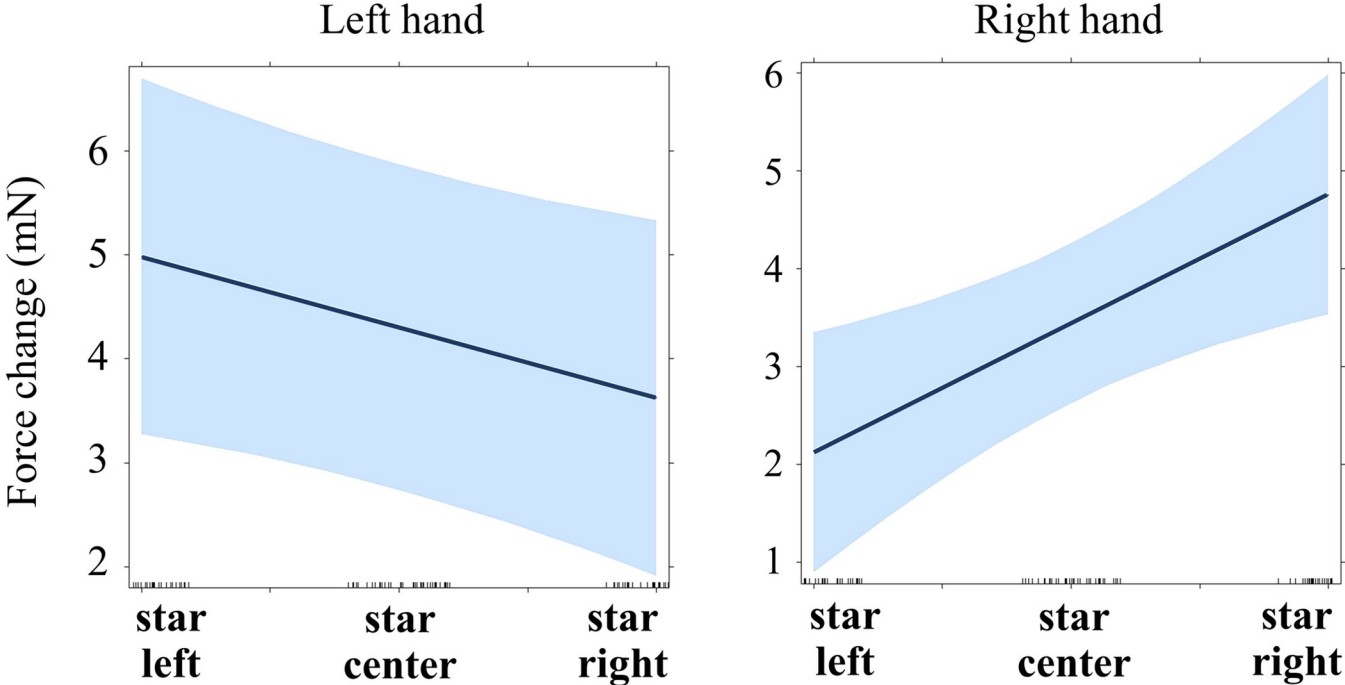

**Fig 4. Regression lines for the main effect of star position on force (Experiment 1, time window 60–130 ms).** See main text for details (Models 1.3 and 1.4).

To summarize, Experiment 1 demonstrated an early (already at 60–130 ms after stimulus onset) interaction between Hand and Position: lateralized stimuli led to relatively stronger force on the ipsilateral side. However, the effect of lateralized stimuli on grip force was rather asymmetric: the effect of Position was more pronounced in the right hand than the left hand. The same pattern emerged at 260–1000 ms, with even higher significance in both hands.

## Experiment 2: Arrow presentation

This experiment induced endogenous attentional shifts by centrally presenting arrows [49, 50] pointing to the left, right, or both directions, while participants' grip force was recorded bimanually.

### Participants

The same sample of participants as in Experiment 1 participated in this experiment.

**Table 4. Model 1.4.** Effect of Position on the right grip force (after controlling for the contralateral force) in the time window 60–130 ms (Experiment 1).

| Random effects: | Name | Variance | SD | |
|---|---|---|---|---|
| Participants | Intercept | 1.083 | 1.041 | |
| Residual | | 17.588 | 4.194 | |
| **Fixed effects:** | **b** | **SE** | **t-value** | **p-value** |
| Intercept | 3.443 | 0.412 | 8.364 | < .001 |
| **Contralateral hand (left)** | **0.493** | **0.063** | **7.811** | < .001 |
| **Position** | **1.316** | **0.464** | **2.839** | .005 |

**Table 5. Model 1.5.** Effect of Position on the left grip force (after controlling for the contralateral force) in the time window 260–1000 ms (Experiment 1).

| Random effects: | Name | Variance | SD | |
|---|---|---|---|---|
| Participants | Intercept | 82.810 | 9.100 | |
| Residual | | 45.420 | 6.740 | |
| **Fixed effects:** | **b** | **SE** | **t-value** | **p-value** |
| Intercept | 1.420 | 1.546 | 0.919 | .358 |
| **Contralateral hand (right)** | **0.217** | **0.089** | **2.436** | **.015** |
| **Position** | **-1.989** | **0.781** | **-2.549** | **.011** |

## Stimuli and design

Red and yellow arrows were used as stimuli in catch (go) and critical (no-go) trials accordingly. The background was kept black. Stimuli can be found in the supplementary materials (see data availability statement). Arrows always appeared in the middle of the screen and pointed to the left, right, or in both directions. Grip force was recorded bimanually. This results in a 2 (Hand: left / right) X 3 (Direction of the arrow: left / both / right) within-participant design.

## Task and procedure

After the calibration procedure already described above, the experiment started. Each trial consisted of a fixation dot (200 ms), followed by a stimulus (until response, but no longer than 2000 ms). Participants saw an arrow around 2 cm in diameter (1.91 degrees of visual angle) with equal probability (33%) pointing into one of three directions (left, right, or both). In 25% of all trials, red arrows appeared, the other 75% of arrows were yellow. The task was to say "yes" when a red arrow appeared. Participants were asked not to cross their legs during the experiment. Critical trials were always no-go trials (yellow arrows). The experiment lasted around 15 minutes and consisted of 360 trials with a break in the middle. A short practice (12 trials) preceded the experiment.

## Data preprocessing and analysis

The same preprocessing procedures were applied as in Experiment 1. One participant had the proportion of trials with force exceeding pre-defined thresholds (±500 mN) larger than 20% and thus was excluded. For the remaining participants, this proportion ranged from 0% to 10% (mean = 2%). Those trials were discarded. Among preserved participants, accuracy varied from 99% to 100% (mean = 100%); error trials were excluded from further analysis.

Force patterns are shown in Fig 6. The solid blue line in Fig 6A represents averaged force of both hands for all accepted no-go trials. Independently of a particular condition, grip force follows a pattern closely resembling that from Experiment 1 with three peaks (H130, H300, and

**Table 6. Model 1.6.** Effect of Position on the right grip force (after controlling for the contralateral force) in the time window 260–1000 ms (Experiment 1).

| Random effects: | Name | Variance | SD | |
|---|---|---|---|---|
| Participants | Intercept | 66.100 | 8.130 | |
| Residual | | 47.340 | 6.881 | |
| **Fixed effects:** | **b** | **SE** | **t-value** | **p-value** |
| Intercept | -1.992 | 1.413 | -1.409 | .159 |
| **Contralateral hand (left)** | **0.220** | **0.086** | **2.545** | **.011** |
| **Position** | **2.949** | **0.770** | **3.832** | **< .001** |

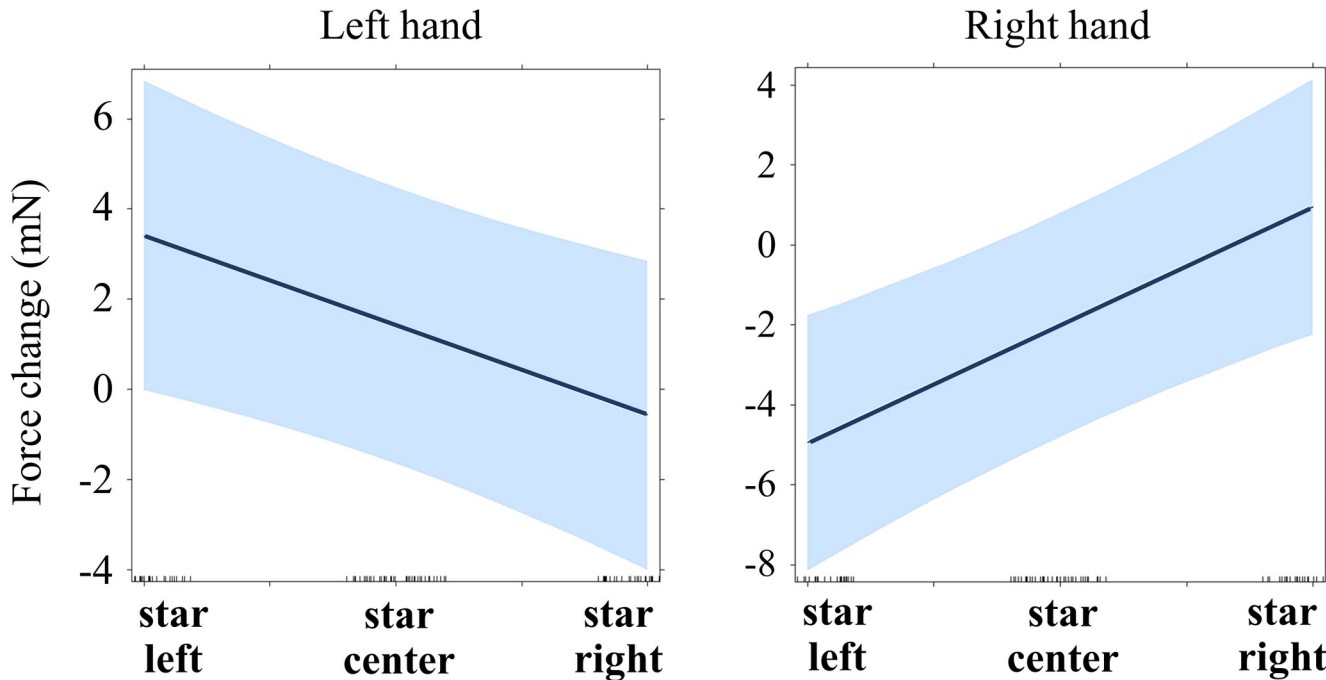

**Fig 5. Regression lines for the main effect of star position on force (Experiment 1, time window 260–1000 ms).** See main text for details (Models 1.5 and 1.6).

H650). This time, the first peak (H130) is the tallest. Fig 6B represents averaged forces in go (dotted red line) and no-go (solid blue line) trials. As in the first Experiment, these two lines start diverging at 250 ms after stimulus onset with force in go trials reaching its highest point (around 55 mN) at around 700 ms and remaining at this level till the end of the epoch (1000 ms). Fig 6C represents force averaged by condition (Hand X Direction).

As before, I performed a cluster permutation analysis with Hand and Direction as within-variables. Five thousand permutations were performed, and TFCE (Threshold-Free Cluster Enhancement) correction for multiple comparisons was used. The analysis revealed a main effect of Direction close to significance in two time windows (510–580 ms and 620–800 ms) and one time window with an interaction between the variables close to significance (100–150 ms; see Fig 7).

**510–580 ms, Model 2.1.** As in Experiment 1, the data (averaged by Hand and Direction for each participant) were then submitted to linear mixed model analysis. The categorical predictor Hand was sum-coded (left/right, sum-coded contrast -0.5 and 0.5). Direction was recoded in a continuous manner (left: -1; both directions: 0; right: 1). Interaction between Hand and Direction was included. Participants were included as random factors. I performed a backward elimination using the drop1 function to identify the best-fit model; effects and interactions that did not improve model fit (p > .1) were successively eliminated. Only significant effects are reported unless the effect of interest in a given analysis was non-significant–in such cases, it is also reported.

No significant effects or interactions were revealed in this time window. The closest to significance was the main effect of Direction, as demonstrated before in the cluster permutation analysis. The force (non-significantly) decreased for right-pointing arrows compared to left-pointing arrows (b = -1.170, p = .100). Marginal r-squared was .007, and conditional r-squared was .384. See Table 7 for further details.

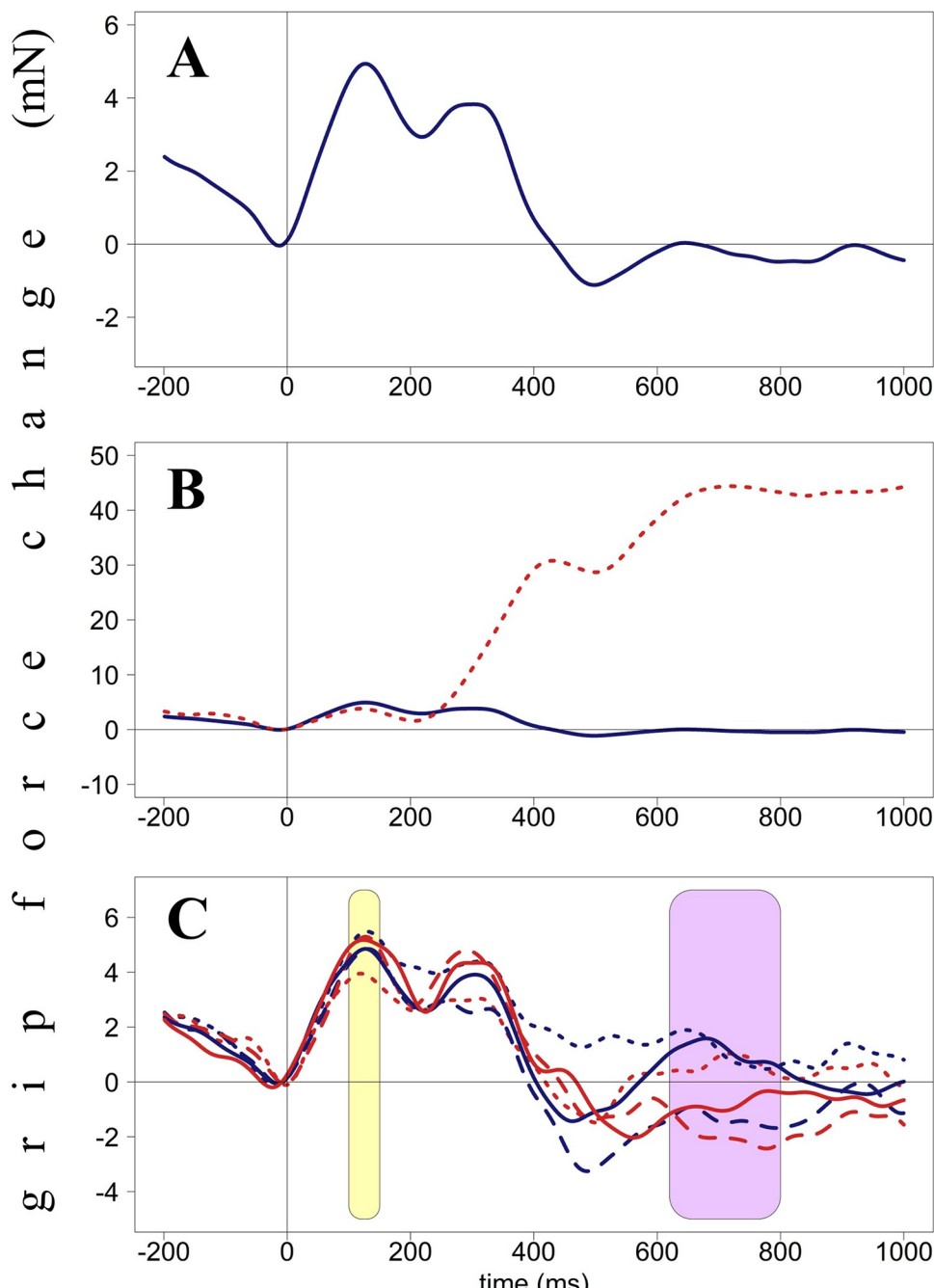

**Fig 6. Grip force changes (in milli-Newton) plotted against time from stimulus onset (in milliseconds), Experiment 2.** Panel A. Averaged force profiles across all accepted no-go trials of all participants. Panel B. Force profiles in go (dotted red line) and no-go (solid blue line) trials. These forces diverge around 250 ms after stimulus presentation. Panel C. Force profiles averaged by condition (Hand X Direction). Yellow area (100–150 ms) indicates interaction between Hand and Direction; violet area (620–800 ms) represents the main effect of Direction close to significance (p < .1). Red lines represent right-hand forces, blue lines–left-hand forces. Dotted lines represent the arrow-left condition, solid lines–arrow-in-both-directions, dashed lines–arrow-right.

**620–800 ms, Model 2.2.** The same approach was applied as in the previous time window. The effect of Direction was marginally significant (p < .1) and thus remained in the model. Again, the force in both hands slightly decreased for right-pointing arrows compared to left-

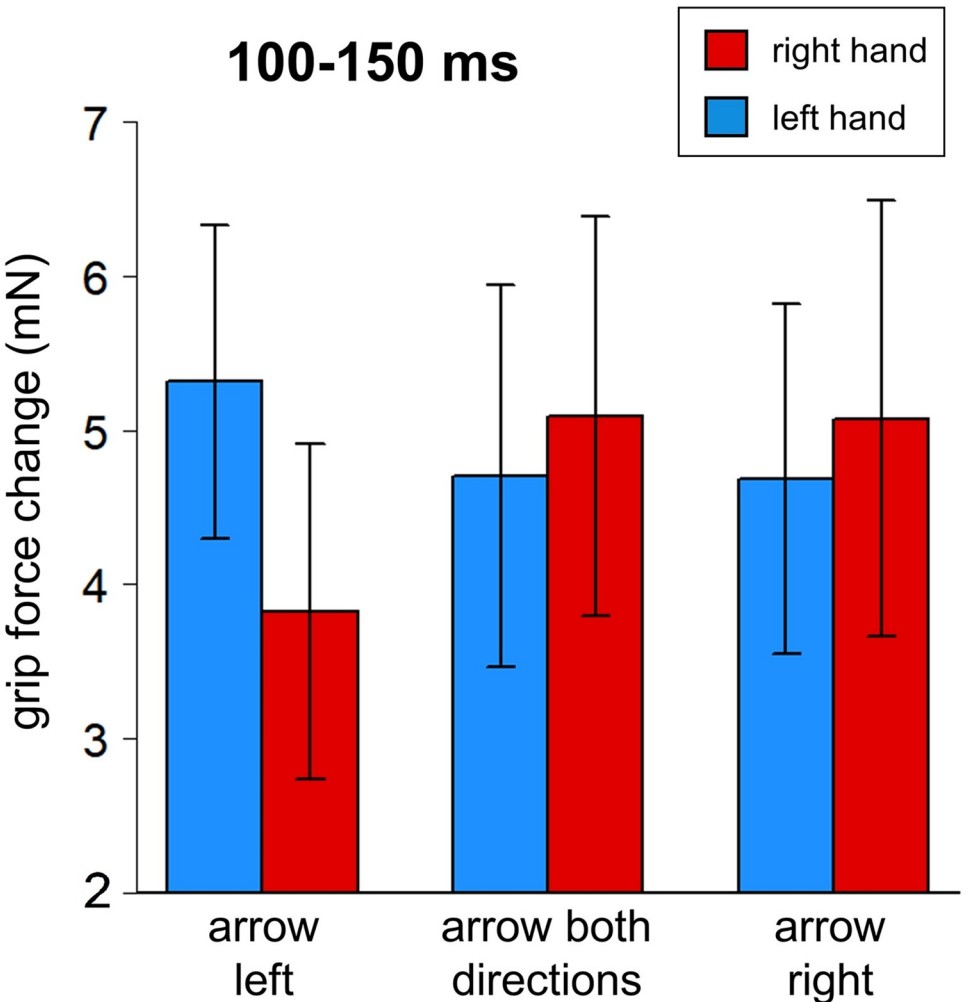

**Fig 7. Grip force changes (in milli-Newton) averaged by Hand and Direction in time widow 100–1350 ms, Experiment 2.** Whiskers represent standard errors.

pointing arrows (b = -1.279, p = .089). The effect of Hand and the interaction between the two variables were not significant. Marginal r-squared was .006, and conditional r-squared was .508. See Table 8 for further details.

**100–150 ms, Models 2.3 and 2.4.** The same approach was used as for Experiment 1 (see time windows 60–130 ms and 260–1000 ms). Direction was re-coded as a continuous variable (left: -1; both directions: 0; right: 1). Each force was tested separately, with the contralateral force and Direction as predictors. The effect of Direction was significant in both hands: the left

**Table 7. Model 2.1.** Main effect of Direction (not significant) on grip force in the time window 510–580 ms (Experiment 2).

| Random effects: | Name | Variance | SD | |
|---|---|---|---|---|
| Participants | Intercept | 49.62 | 7.044 | |
| Residual | | 80.90 | 8.994 | |
| **Fixed effects:** | **b** | **SE** | **t-value** | **p-value** |
| Intercept | -0.772 | 1.256 | -0.615 | 0.539 |
| **Direction** | **-1.170** | **0.711** | **-1.645** | **0.100** |

**Table 8. Model 2.2.** Main effect of Direction on grip force in the time window 620–800 ms (Experiment 2).

| Random effects: | Name | Variance | SD | |
|---|---|---|---|---|
| Participants | Intercept | 92.23 | 9.604 | |
| Residual | | 90.25 | 9.500 | |
| **Fixed effects:** | **b** | **SE** | **t-value** | **p-value** |
| Intercept | -0.211 | 1.638 | -0.129 | .898 |
| **Direction** | **-1.279** | **0.751** | **-1.702** | **.089** |

force increased when stimuli arrows pointed to the left (b = -0.717, p = .028; marginal r-squared = .599, conditional r-squared = .810; see Table 9) and the right force increased when the arrows pointed to the right (b = 0.867, p = .013; marginal r-squared = .566, conditional r-squared = .817; see Table 10; see also Fig 8).

Thus, Experiment 2 demonstrated a pattern similar to Experiment 1: left- and right-pointing arrows led to a significant force decrease in the contralateral hand. Unlike in Experiment 1, where the effect was stronger in the right hand, the magnitude of the effect in Experiment 2 was comparable across hands. Moreover, the effect of arrow direction was less pronounced and appeared later than the effect of star position.

## Experiment 3: Word presentation

In Experiment 3, participants saw centrally presented words LINKS, RECHTS or ZENTRUM ("left", "right" or "center" in German; Note, however, that words "LINKS" and "RECHTS" are adverbs in German, while the word "ZENTRUM" is a noun). Bimanual force recording allowed to investigate dynamic involvement of the motor system into the processing of spatial information presented in a purely symbolic way, i.e., through linguistic meaning.

### Participants

Only a subsample of German native speakers participated in this experiment (N = 27; mean age = 24; 9 males and 18 females). On average, participants spoke 1.85 foreign languages, most frequently English, Spanish, and French. The mean EHI score of those participants was +60, with 21 participants (78%) having EHI score > +50, 2 participants (7%) having EHI scores between +50 and -50, and 4 participants (15%) with EHI score < -50. All participants reported normal or corrected-to-normal vision. No participant took medications affecting motor control.

### Stimuli and design

Red and yellow words or meaningless symbol arrays (e.g., $@#$%) were used as stimuli in catch (go) and critical (no-go) trials accordingly. Symbol arrays were included to investigate

**Table 9. Model 2.3.** Effect of Direction on the left grip force (after controlling for the contralateral force) in the time window 100–150 ms (Experiment 2).

| Random effects: | Name | Variance | SD | |
|---|---|---|---|---|
| Participants | Intercept | 9.364 | 3.060 | |
| Residual | | 8.412 | 2.900 | |
| **Fixed effects:** | **b** | **SE** | **t-value** | **p-value** |
| Intercept | 4.900 | 0.552 | 8.884 | < .001 |
| **Contralateral hand (right)** | **0.644** | **0.058** | **11.080** | **< .001** |
| **Direction** | **-0.717** | **0.326** | **-2.199** | **.028** |

**Table 10. Model 2.4.** Effect of Direction on the right grip force (after controlling for the contralateral force) in the time window 100–150 ms (Experiment 2).

| Random effects: | Name | Variance | SD | |
|---|---|---|---|---|
| Participants | Intercept | 13.303 | 3.647 | |
| Residual | | 9.688 | 3.112 | |
| **Fixed effects:** | **b** | **SE** | **t-value** | **p-value** |
| Intercept | 4.663 | 0.643 | 7.253 | < .001 |
| **Contralateral hand (left)** | **0.767** | **0.072** | **10.638** | **< .001** |
| **Direction** | **0.867** | **0.349** | **2.486** | **.013** |

potential lexicality effects, i.e., the difference in signal between word and non-word stimuli found, for example, in EEG at 150–250 ms after stimulus onset [51]. The background was kept black. Experimental scripts can be found in the supplementary data (see data availability statement). Grip force was recorded bimanually. This results in a 2 (Hand: left / right) X 3 (Word: left / center / right) within-participant design.

## Task and procedure

After the calibration procedure described above, the experiment started. Each trial consisted of a fixation dot (200 ms), followed by a stimulus (until response, but no longer than 2000 ms). Participants saw words or symbol arrays, all having a length of around 2.5 cm (2.39 degrees of visual angle) in a 25 px (19 pt) Droid Sans Mono font. Words "left", "right", and "center" appeared with equal probability. The task was to say "yes" when a red word (20% of all trials) or symbol array (20% of all trials) appeared. Participants were asked not to cross their legs during the experiment. Critical trials were always no-go with words (i.e., only real words in yellow font, 40% of all trials). In the remaining 20% of all trials, yellow symbol arrays appeared, and

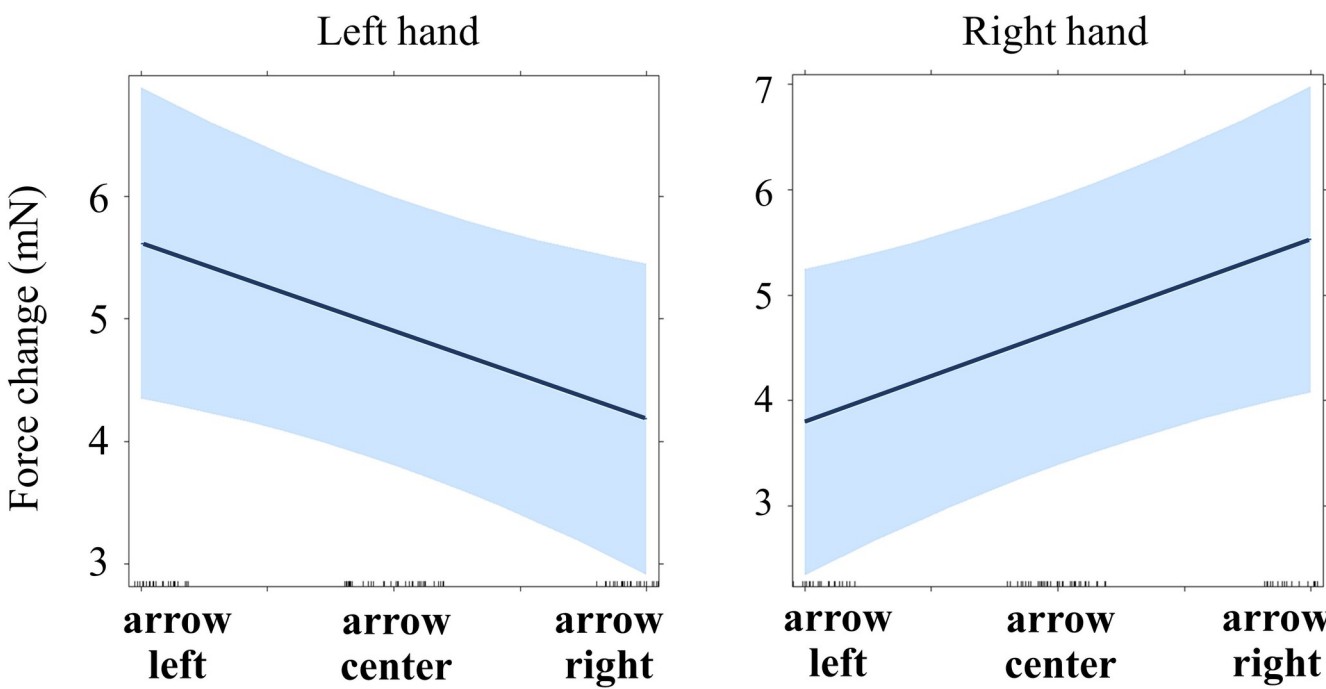

**Fig 8. Regression lines for the main effect of arrow direction on force (Experiment 2, time window 100–150 ms).** See main text for details (Models 2.3 and 2.4).

no response was required. The experiment consisted of 450 trials with a break in the middle and lasted around 20 minutes. A short practice (18 trials) preceded the experiment.

### Data preprocessing and analysis

The same preprocessing procedures were applied as in Experiment 1. The proportion of trials with force exceeding pre-defined thresholds (±500 mN) ranged from 0% to 18% (mean = 2%), and no participant was excluded due to this criterion. Those trials were discarded. Accuracy varied from 97% to 100% (mean = 99%); error trials were excluded from further analysis.

Fig 9A represents averaged force of both hands for all accepted no-go trials plotted for words (blue line) vs. symbol arrays (pink line). Independently of condition, grip force demonstrates the following pattern: there are two well-defined peaks of equal height (H130 and H350) with a slight deviation of force at the beginning of the second peak (H250). After H350, the force drops dramatically until 600 ms and remains relatively stable until the end of the epoch (1000 ms), with only a tiny wave having its peak at H850. Note that the pattern for symbol arrays closely resembles that for words, especially in the critical time window around 200 ms. The only difference between the two forces is the larger peak for symbol arrays around 650 ms. Fig 9B represents averaged forces in go (dotted red line) and no-go (solid blue line) trials for words. These two lines start diverging at 230 ms after stimulus onset with force in go trials reaching its highest point (around 40 mN) at around 650 ms and remaining at this level till the end of the epoch. Fig 9C represents force averaged by condition (Hand X Word).

As before, a cluster permutation analysis was used for exploratory purposes. First, Hand (right / left) and Lexicality (word / symbol array) as within-variables and their interaction were submitted. The analysis did not reveal any significant effects (all p-values > .72). This result indicates that lexicality is not reflected in the force signal; trials with symbol arrays were excluded from all further analyses. Next, a cluster permutation analysis was performed with Hand and Word as within-variables and their interaction. Five thousand permutations were performed, and TFCE (Threshold-Free Cluster Enhancement) correction for multiple comparisons was used. The analysis revealed a main effect of Hand close to significance (880–970 ms) and two time windows with an interaction between Hand and Word close to significance (250–300 ms, see Fig 10, and 810–890 ms).

**880–970 ms, Model 3.1.** As in Experiments 1 and 2, the data (averaged by Hand and Word for each participant) were then submitted to linear mixed model analysis. The categorical predictor Hand was sum-coded (left/right, sum-coded contrast -0.5 and 0.5). Word was re-coded in a continuous manner (left: -1; center: 0; right: 1). Interaction between Hand and Word was included. Participants were included as random factors. I performed a backward elimination using the drop1 function to identify the best-fit model; effects and interactions that did not improve model fit (p > .1) were successively eliminated. Only significant effects are reported unless the effect of interest in a given analysis was non-significant–in such cases, it is also reported.

No significant effects or interactions were revealed in this time window. The closest to significance was the main effect of Hand. The force was (non-significantly) lower in the left hand compared to the right hand (b = -3.355, p = .097). Marginal r-squared was .013, and conditional r-squared was .249. See Table 11 for further details.

**250–300 ms, Models 3.2 and 3.3.** The same approach was used as for Experiment 1 (see time windows 60–130 ms and 260–1000 ms). Word was re-coded as a continuous variable (left: -1; center: 0; right: 1). Each force was tested separately, with the contralateral force and Word as predictors. The effect of Word was significant in the left hand: the grip force increased for the word "right" compared to the word "left" (b = 1.736, p = .047; marginal r-squared =

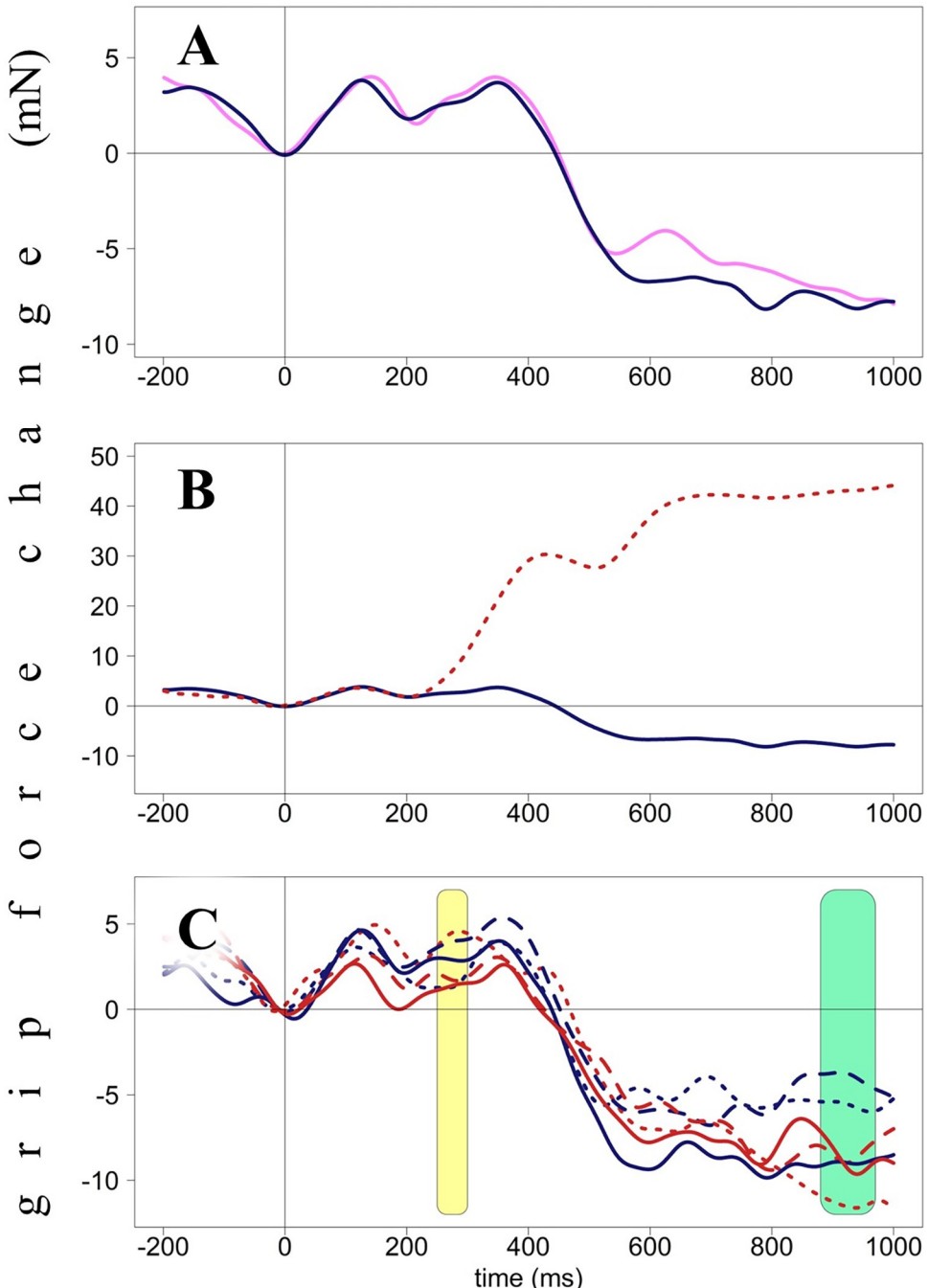

**Fig 9. Grip force changes (in milli-Newton) plotted against time from stimulus onset (in milliseconds), Experiment 3.** Panel A. Averaged force profiles across all accepted no-go trials of all participants plotted for words (blue line) vs. symbol arrays (pink line). Panel B. Force profiles in go (dotted red line) and no-go (solid blue line) trials. These forces diverge around 230 ms after stimulus presentation. Panel C. Force profiles averaged by condition (Hand X Word). Yellow area (250–300 ms) indicates interaction between Hand and Word. Green area (880–970 ms) indicates the main effect of Hand (not significant). Red lines represent right-hand forces, blue lines–left-hand forces. Dotted lines represent the word-left condition, solid lines–word-center, dashed lines–word-right.

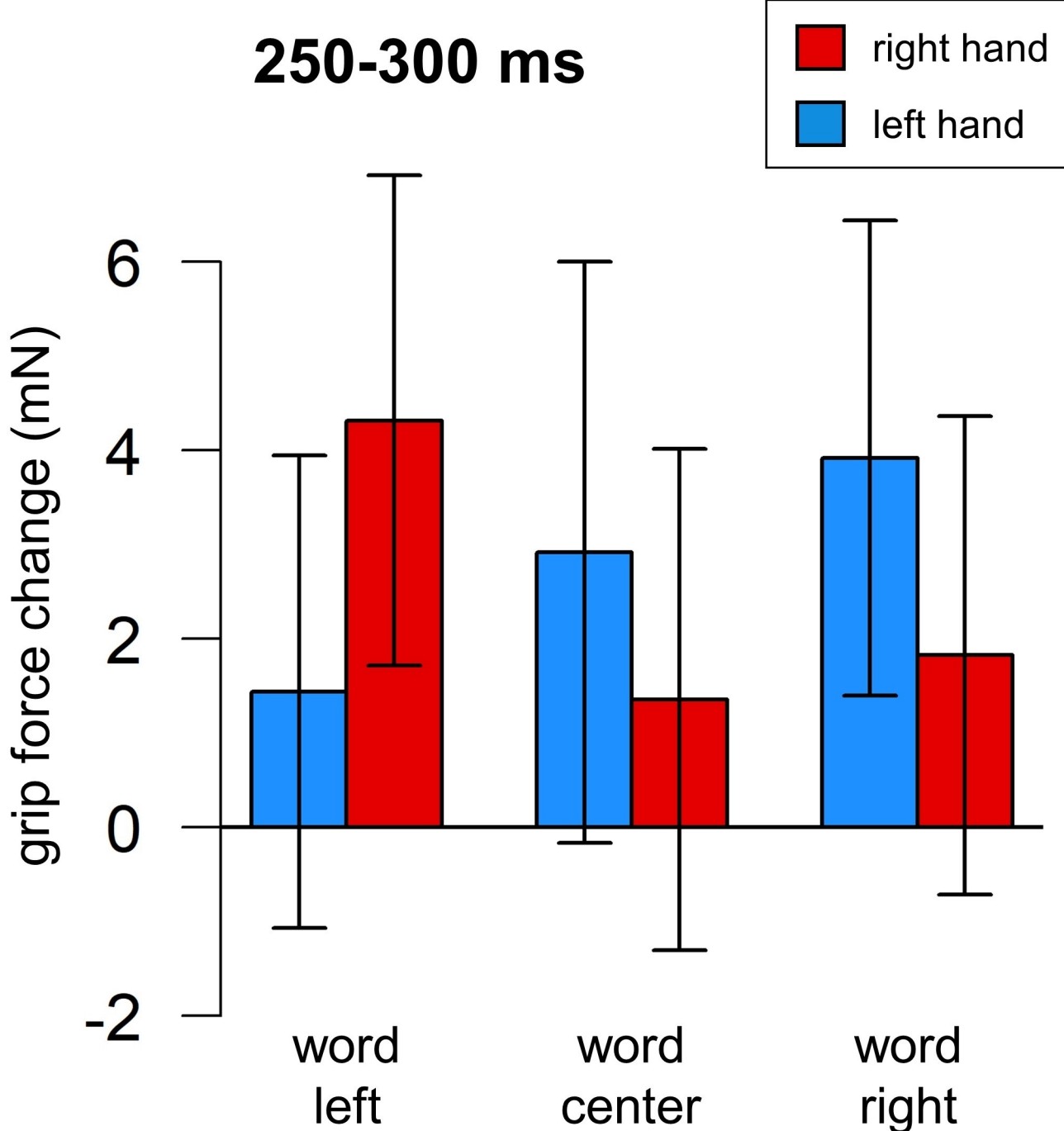

**Fig 10. Grip force changes (in milli-Newton) averaged by Hand and Word in time widow 250–300 ms, Experiment 3.** Whiskers represent standard errors.

.171, conditional r-squared = .764; see Table 12), and in the right hand the force increased for the word "left" compared to the word "right" (b = -1.690, p = .041; marginal r-squared = .166, conditional r-squared = .768; see Table 13; see also Fig 11).

**810–890 ms, Models 3.4 and 3.5.** The same approach was used as for the previous time window. The effect of Word was not significant in the left hand (b = 0.121, p = .938; marginal

**Table 11. Model 3.1.** Main effect of Hand on grip force in the time window 880–970 ms (not significant; Experiment 3).

| Random effects: | Name | Variance | SD | |
|---|---|---|---|---|
| Participants | Intercept | 52.2 | 7.225 | |
| Residual | | 165.9 | 12.882 | |
| **Fixed effects:** | **b** | **SE** | **t-value** | **p-value** |
| Intercept | -7.866 | 1.720 | -4.574 | .001 |
| **Hand** | **-3.355** | **2.024** | **-1.657** | **.097** |

r-squared = .210, conditional r-squared = .501; see Table 14), neither was it significant in the right hand (b = 0.638, p = .619; marginal r-squared = .188, conditional r-squared = .596; see Table 15).

Thus, the effect of Word emerged at 250–300 ms, which is later than the effect of Position (Experiment 1) or Direction (Experiment 2). Moreover, the grip force increased in hand contralateral to the expected one: in the right hand for the word "left" and in the left hand for the word "right". This surprising finding will be discussed in detail in the last section.

## Analysis at the individual level across experiments

Since the same individuals participated in all three experiments, it was possible to analyze data across studies. Note that this analysis is rather exploratory due to the small number of participants. I added random slopes to all models where significant interaction between Hand and Side (Position, Direction, or Word) emerged, for left and right hands separately: 60–130 ms and 260–1000 ms (Experiment 1), 100–150 ms (Experiment 2), and 250–300 ms (Experiment 3). These individual coefficients were submitted into a correlational analysis (see Table 16).

## Correlations between hands in the same time windows within the same experiments

A negative significant correlation (r = −.438, p = .004) was found between the right-hand and left-hand coefficients in the time window 260–1000 ms (Experiment 1, star presentation). Since the effect of Side is oppositely directed in the left and right hands, this negative correlation demonstrates that both forces were influenced by the stimuli simultaneously at the individual level. In other words, the same participants, who demonstrated the effect in one hand, were also more likely to demonstrate it in another hand. The same pattern was found for the time window 100–150 ms in Experiment 2 (arrow presentation; r = −.402, p = .010). The correlation for the time window 60–130 ms in Experiment 1 (star presentation) was close to significance (r = −.268, p = .090). The correlation for the time window 250–300 ms in Experiment 3 (word presentation) was also negative, though far from significance (r = .217, p = .288). Overall, these results show that Position and Direction have simultaneous effects on both hands at

**Table 12. Model 3.2.** Effect of Direction on the left grip force (after controlling for the contralateral force) in the time window 250–300 ms (Experiment 3).

| Random effects: | Name | Variance | SD | |
|---|---|---|---|---|
| Participants | Intercept | 101.23 | 10.061 | |
| Residual | | 40.39 | 6.355 | |
| **Fixed effects:** | **b** | **SE** | **t-value** | **p-value** |
| Intercept | 2.761 | 2.061 | 1.339 | .180 |
| **Contralateral hand (right)** | **0.397** | **0.108** | **3.680** | **< .001** |
| **Word** | **1.736** | **0.875** | **1.983** | **.047** |

**Table 13. Model 3.3.** Effect of Direction on the right grip force (after controlling for the contralateral force) in the time window 250–300 ms (Experiment 3).

| Random effects: | Name | Variance | SD | |
|---|---|---|---|---|
| Participants | Intercept | 94.34 | 9.713 | |
| Residual | | 36.25 | 6.021 | |
| **Fixed effects:** | **b** | **SE** | **t-value** | **p-value** |
| Intercept | 2.502 | 1.985 | 1.260 | .208 |
| **Contralateral hand (left)** | **0.358** | **0.098** | **3.648** | **< .001** |
| **Word** | **-1.690** | **0.828** | **-2.041** | **.041** |

the individual level. For Word, this cannot be concluded, perhaps due to absence of the effect or a smaller number of participants in Experiment 3 (N = 26).

## Correlations across time windows within the same experiment

More interesting is the positive correlation between right-hand slope coefficients for the time window 60–130 ms and 260–1000 ms (Experiment 1, star presentation; r = .625, p < .001). This correlation suggests that the effect of Position observed in the earlier time window (60–130 ms) and in the later one (260–1000 ms) is substantially the same effect that was interrupted between 130–260 ms, probably due to large force oscillations (H130 and L230, see Experiment 1). No such correlation was observed for the left-hand coefficients (r = −.013, p = .936).

## Correlations across experiments

Right-hand coefficients in the time window 60–130 ms, Experiment 1, correlated positively with right-hand coefficients in the time window 100–150 ms, Experiment 2 (r = .479, p =

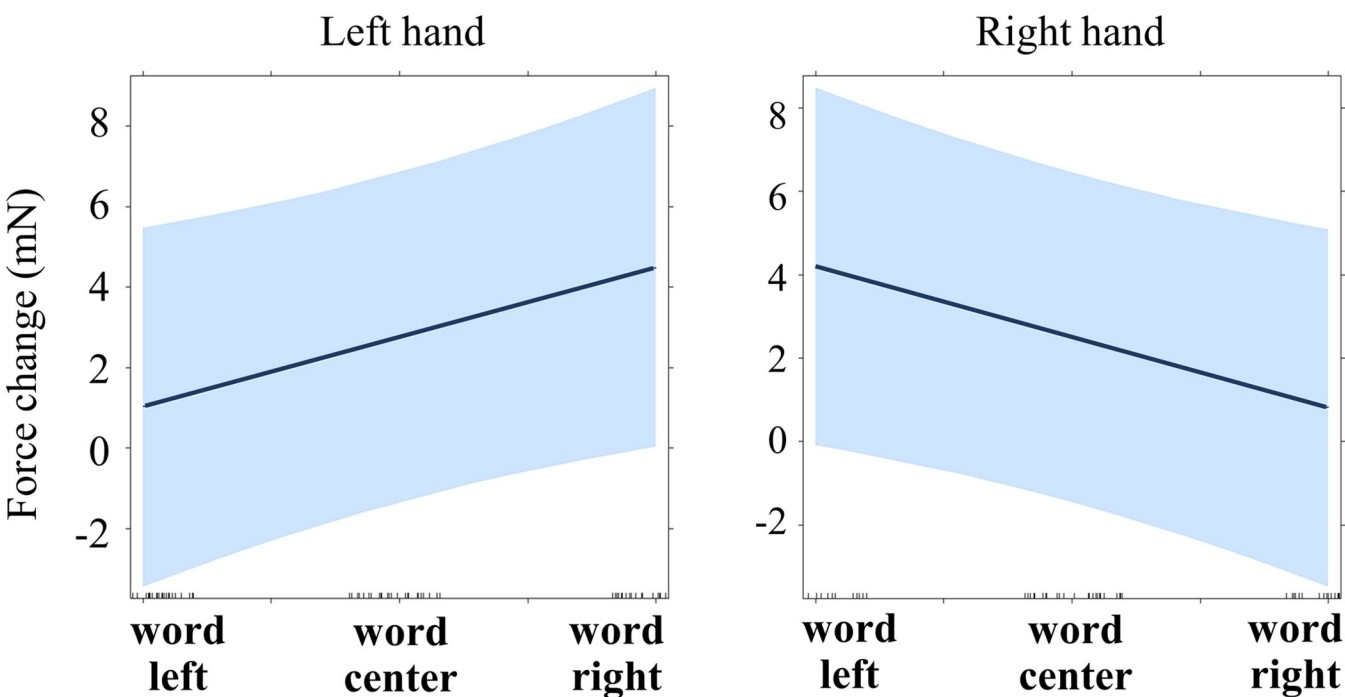

**Fig 11. Regression lines for the main effect of word semantics on force (Experiment 3, time window 250–300 ms).** See main text for details (Models 3.2 and 3.3).

**Table 14. Model 3.4.** Effect of Direction on the left grip force (after controlling for the contralateral force) in the time window 810–890 ms (Experiment 3).

| Random effects: | Name | Variance | SD | |
|---|---|---|---|---|
| Participants | Intercept | 75.94 | 8.715 | |
| Residual | | 130.15 | 11.408 | |
| **Fixed effects:** | **b** | **SE** | **t-value** | **p-value** |
| Intercept | -6.270 | 2.102 | -2.982 | .003 |
| **Contralateral hand (right)** | **0.506** | **0.112** | **4.523** | **< .001** |
| **Word** | **0.121** | **1.555** | **0.078** | **.938** |

.002). A negative correlation was observed for left-hand coefficients in the time window 60–130 ms, Experiment 1, and right-hand coefficients in the time window 100–150 ms, Experiment 2 (r = –.495, p = .001). These correlations demonstrate that the same participants, who showed the effect of Position in Experiment 1 (star presentation), were more likely to exhibit a comparable effect of Direction in Experiment 2 (arrow presentation). More surprising are the other two negative correlations: the one between right-hand coefficients at 60–130 ms in Experiment 1 and at 250–300 ms in Experiment 3 (r = –.435, p = .026); and the other one between left-hand coefficients at 260–1000 ms in Experiment 1 and at 250–300 ms at Experiment 3 (–.510, p = .008). These two correlations indicate that the same participants who demonstrated the effect of Position in Experiment 1 (star presentation) also demonstrated the effect of Word in Experiment 3, although the effect of Word was the opposite of the expected.

In the next section, all results will be discussed in more detail.

## Discussion

The present study aimed to investigate the effects of spatial processing on the manual motor system by using a new method–bimanual grip force recording. In Experiment 1, participants' visual attention was shifted using lateralized stimuli presentation (Experiment 1). In Experiment 2, participants were centrally presented with pictographic symbols with spatial meaning (left- or right-oriented arrows). In Experiment 3, participants were centrally presented with words having spatial meaning ("left" vs. "right"). Since a go/no-go paradigm with a verbal response in go trials was used, any observed effects can only be attributed to spatial/semantic processing alone and not to motor preparation of responses.

### General pattern of grip force changes

The first important finding is that all three types of stimuli (stars, arrows, and words) led to very similar initial force patterns (see Fig 12), namely two peaks (around 130 and 300–350 ms after stimulus onset) followed by differentially declining force profiles. I will now interpret these results in turn.

**Table 15. Model 3.5.** Effect of Direction on the right grip force (after controlling for the contralateral force) in the time window 810–890 ms (Experiment 3).

| Random effects: | Name | Variance | SD | |
|---|---|---|---|---|
| Participants | Intercept | 89.61 | 9.466 | |
| Residual | | 88.60 | 9.413 | |
| **Fixed effects:** | **b** | **SE** | **t-value** | **p-value** |
| Intercept | -8.538 | 2.101 | -4.064 | < .001 |
| **Contralateral hand (left)** | **0.408** | **0.086** | **4.762** | **< .001** |
| **Word** | **0.638** | **1.282** | **0.498** | **.619** |

**Table 16. Correlations between individual slopes across three experiments.**

|  | S 60–130 RH | S 260–1000 LH | S 260–1000 RH | A 100–150 LH | A 100–150 RH | W 250–300 LH | W 250–300 RH |
|---|---|---|---|---|---|---|---|
| N | 41 | 41 | 41 | 40 | 40 | 26 | 26 |
| S 60–130 LH | − .268 | − .013 | − .030 | − .300 | **− .495**\*\* | .250 | .174 |
|  | p = .090 | p = .936 | p = .853 | p = .061 | **p = .001** | p = .218 | p = .394 |
|  |  |  |  |  |  |  |  |
| S 60–130 RH |  | **− .394**\* | **.625**\*\*\* | .070 | **.479**\*\* | .257 | **− .435**\* |
|  |  | **p = .011** | **p < .001** | p = .670 | **p = .002** | p = .205 | **p = .026** |
| S 260–1000 LH |  |  | **− .438**\*\* | .147 | − .286 | **− .510**\*\* | .015 |
|  |  |  | **p = .004** | p = .366 | p = .074 | **p = .008** | p = .944 |
| S 260–1000 RH |  |  |  | .253 | .174 | .230 | − .278 |
|  |  |  |  | p = .115 | p = .283 | p = .259 | p = .170 |
| A 100–150 LH |  |  |  |  | **− .402**\* | − .289 | − .111 |
|  |  |  |  |  | **p = .010** | p = .152 | p = .591 |
| A 100–150 RH |  |  |  |  |  | .197 | − .177 |
|  |  |  |  |  |  | p = .336 | p = .388 |
| W 250–300 LH |  |  |  |  |  |  | − .217 |
|  |  |  |  |  |  |  | p = .288 |

Numbers in variable names denote time windows. S–stars (Experiment 1); A–arrows (Experiment 2); W–words (Experiment 3). LH–left-hand force; RH–right-hand force. N–number of participants included in the analysis (varied across experiments).

\*p < .05

\*\*p < .01

\*\*\*p < .001.

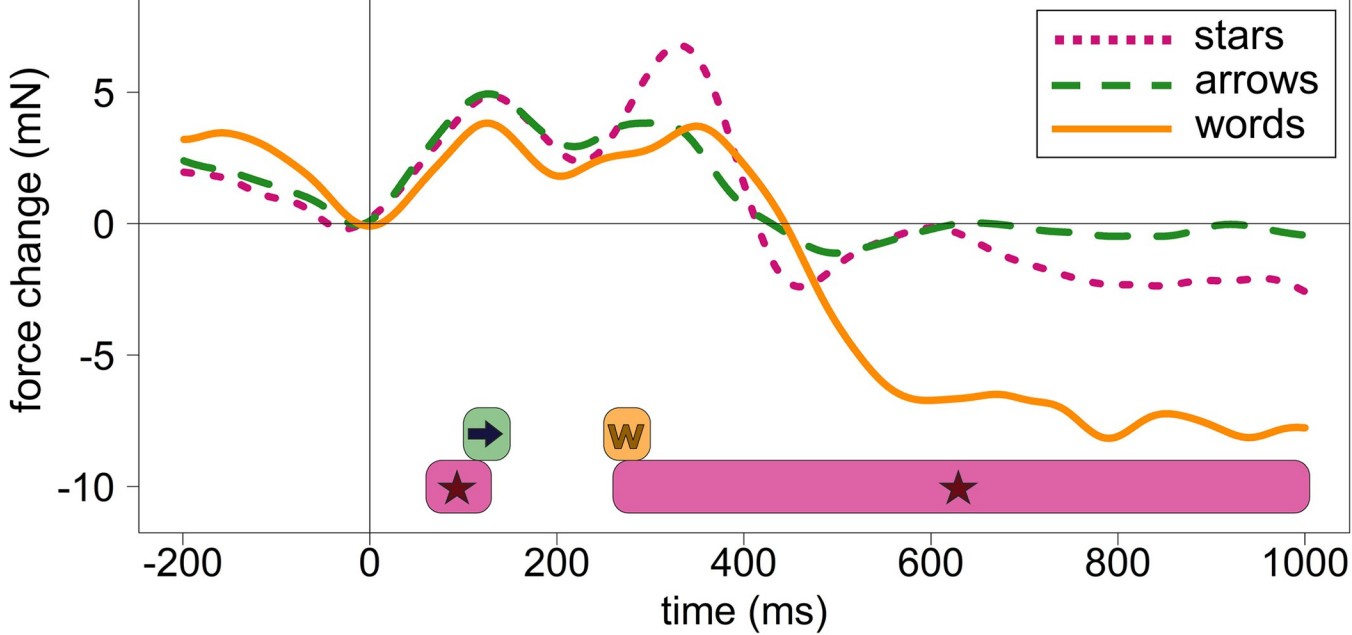

**Fig 12. Averaged forces from both hands across all no-go trials (regardless of condition) in three experiments.** The dotted pink line represents force modulations caused by the presentation of stars (Experiment 1), dashed green line–arrows (Experiment 2), solid orange line–words (Experiment 3). Colored horizontal bars in the lower part of the graph represent time windows with significant Side X Hand interactions for all three types of stimuli–stars, arrows, and words. The colors of bars correspond to the colors of lines described above.

This pattern is very close to force profiles observed in studies with other stimuli–numbers [15] and human faces (Miklashevsky et al., in prep.). In response to numbers, grip force peaked at 100 ms and later at 350 ms. Two more prominent peaks (around 100–130 and 300–350 ms) probably reflect two processing stages: (1) initial processing of any stimulus appearing on a screen and (2) decision-making or response inhibition process accordingly. Remember that only no-go trials are included in analyses. The divergence point between forces (230–250 ms) in go and no-go trials always precedes the second peak–i.e., at this point, participants' arousal increases in trials requiring response and should be inhibited in trials where no response is needed. These assumptions require confirmation in future studies.

Interestingly, a slight deviation formed an additional peak at 250 ms in an earlier study with numbers [15] which results in a pattern very similar to those observed for words and (less pronounced) arrows in the present study but not for stars (in this study) or faces (in a previous study, unpublished). Since numbers, words, and arrows are all symbolic stimuli, I hypothesize that this smaller peak around 250 ms reflects some process specific to the understanding of symbols.

Suppose my hypothesis is correct and grip force reflects not only motor-related but also more general cognitive processes, such as stimulus identification and preparation or inhibition of verbal responses. In that case, this has implications for other studies using the force registration method. In previous studies of this kind [e.g., 19, 21], force changes were always interpreted as specific signatures of activity in the manual motor system. However, a recent study shows that even observing foot actions leads to changes in grip force [30], perhaps due to automatic propagation of activity in the motor brain areas. The same principle might be at play in the present study. Thus, it might be methodologically incorrect to choose force at the time point of stimulus onset as a baseline and compare different conditions with it since any stimulus might lead to force oscillations. Different conditions should be compared with each other instead, or a continuous coding of variables should be chosen as in the present study.

### Influence of spatial processing on grip force changes

Regarding the specific hypothesis of the present study, i.e., whether or not spatial attentional shifts lead to the automatic activation of the manual motor system–the answer is yes, although with some reservations. When lateralized stimuli appeared (stars, Experiment 1), a significant difference in force emerged already at 60–130 ms and later at 260–1000 ms. The force increased larger in each of the hands when the star was presented on the ipsilateral side; this increase was smaller for stars presented on the contralateral side. This effect was stronger in the right hand in both time windows. Since individual linear coefficients for both time windows (60–130 ms and 260–1000 ms) correlated positively for the right hand, I suppose these two time windows reflect the same effect. The interruption between the two windows appears due to the multiphasic structure of force profiles: forces in all conditions go down rapidly between 130 and 230 ms, thus reducing differences between conditions. Note that no significant effects were found in all three experiments in this time interval.

A similar pattern, although less pronounced, appeared for arrows (Experiment 2): the force increased larger in each of the hands for arrows pointing in that hand's direction; the increase was smaller for arrows pointing in the opposite direction. This effect emerged later than (at 100 ms), lasted shorter (just 50 ms), and was weaker than for stars. Nevertheless, it was also found in the first "wave" (H130), i.e., it probably belongs to the same processing stage. It is not surprising since symbolic cues have a similar impact on attention as physical ones, even if these symbolic cues are task-irrelevant [see 52, 53]. The present study provides even more substantial evidence in favor of the automatic effect of arrows on the attentional system than the

studies by Hommel, Pratt et al. [52], or Ranzini et al. [53]. In these previous studies, the arrows themselves were non-predictive, but lateralized stimuli still followed arrows. In the present study, no lateralized stimuli were used (Experiment 2), yet force patterns resembled those produced by exogenous attentional shifts. The effect for arrows (Experiment 2), measured as regression coefficients, correlated with the effect for stars (Experiment 1) at the individual level, indicating stable inter-individual differences in motor response to spatial stimuli.

Arrows are viewed as symbolic stimuli in the present study, although there is a debate about their actual status: arrows demonstrate effects typical for both exogenous and endogenous cues. Ristic and Kingstone even speak of the third form of attention–automated symbolic orienting [34]. In the present study, the general force pattern for arrows is similar to both: the pattern produced by words (the second peak, H300, is lower than the first one, H130), but also to the pattern produced by stars (grip force does not drop as deeply as for words after 300–350 ms and there is a slight increase around 600–650 ms). The same is true regarding the effect of spatial information: qualitatively similar to those produced by stars, it is at the same time weaker for arrows. It appears later than for stars but earlier than for words.

A significant effect of word semantics on force was found in Experiment 3. Yet, the direction of this effect was surprising: the force increased more for words related to the opposite side; this increase was smaller for words related to the same side. The effect appeared rather late (250–300 ms) and was of comparable strength across both hands. Note that it belongs to the second larger force wave and emerges simultaneously with the tiny force deviation discussed above (H250). This information might be crucial for interpreting the direction of the effect: remember that participants inhibited their verbal response in no-go trials, and the divergence point between go and no-go trials, presumably reflecting the moment of such inhibition, was exactly 230 ms. The stronger H250 wave probably results from the conflict between the two overlapping processes: automatic activation of word semantics and conscious inhibition of verbal response. By suppressing semantics of the word "left", participants literally had to inhibit activity in the left hand [for more details on the interplay between the semantic and motor system see the HANDLE model, 54]; the same happened with the word "right" and the right hand. To further examine this hypothesis, it would be necessary to design a study following the procedure of Experiment 3 but with a non-linguistic response. I predict that words will exhibit force patterns similar to those in Experiments 1 and 2 in such a study since no inhibition of motor semantics should happen. If this is the case, this finding has implications for the research on inhibitory control: multiple measures of this presumably higher-order construct often do not correlate with each other [e.g., 55]. The reason might be that inhibitory control does not constitute a single construct but is instead highly domain-specific.

An additional finding in Experiment 3 is that lexicality, i.e., the distinction between word and non-word stimuli, previously found to influence the EEG signal at 150–250 ms after stimulus onset [51], is not reflected in grip force signal. Unlike in most studies on lexical processing, meaningless symbol arrays (§@#$%) were used in the present study instead of pseudowords. Future research should consider using pseudowords built according to phonetic and morphological regularities of the language under investigation to confirm the present finding.

The findings of the present study are in line with neuroscientific research of attention. ERP research established a fine-graded time course of attentional processes. Attentional effects on the processing of lateralized stimuli were found in the magnitude of P1 (60–100 ms after stimulus onset), N1 (around 150 ms), and P2 (around 200 ms) ERP components [see for review 56]. Effects of symbolic control of attention induced by using centrally presented arrows appear at 200–400 ms after cue onset over contralateral posterior sites (the so-called early directing attention negativity, EDAN), probably reflecting encoding spatial information

provided by the cue. EDAN follows by anterior directing attention negativity (ADAN) appearing over contralateral anterior sites at 300–500 ms and performing attentional shift. Finally, a positive waveform (late directing attention positivity, LDAP) appears after 500 ms after cue onset and presumably represents top-down modulation in the excitability of sensory areas [also 53, see for review 57]. This observation aligns with reaction time research demonstrating the highest performance in detecting targets around 300 ms after the onset of a symbolic cue. Still, this effect appears even earlier [see 58, for discussion]. The present study confirms very early effects of both lateralized visual stimuli (60–130 ms) and centrally presented symbolic cues (100–150 ms). It also demonstrates that these effects are detectable in the manual motor system. Such early and automatic modulation is one of the signatures of functional coupling between two structures and not mere spreading spillover activation [see 59].

Although the present study shares many similarities with classical research using the Posner paradigm, substantial differences in the setup and procedure should be considered when comparing the current results with previous studies. First, no laterally appearing stars, arrows, or words were cues in the original sense of this term: there were no "target" stimuli following them. Instead, they were themselves the targets. While lateralized stars indeed led to automatic attentional shifts due to their location, symbolic cues (arrows and words) could indirectly shift attention, following automatic processing of their meaning, which was not necessary for the color discrimination task. The task itself required merely superficial processing [60] and was not related to spatial information. These factors taken together make the similarity between effects of lateralized stimuli and centrally presented arrows even more remarkable. Further research should vary the role of stimuli (by turning them into cues with varying validity as in the original Posner paradigm) and complexity of the motor task to clarify the exact functional relationship between the attentional and manual motor systems [cf. 61].

## Conclusion

The present study investigated the relationship between spatial attention and the manual motor system by using a go/no-go paradigm and bimanual registration of grip force. Automatic and rapid changes in grip force were found in response to lateralized visual stimuli (Experiment 1) and centrally presented symbolic stimuli (Experiments 2, arrows, and 3, words). This activity followed a similar early biphasic pattern for all kinds of stimuli: one peak emerged at 130 ms and another at 300–350 ms after stimulus onset. For both lateralized objects and centrally presented arrows, the direction of the effect was as predicted, i.e., the left force increased in response to objects presented on the left side or left-pointing arrows, while the opposite was true for the right force. This effect appeared very early for lateralized objects (60 ms) and slightly later for arrows (100 ms). A reverse pattern was observed for words: each of the two forces increased for words related to the opposite side. The effect for words was significant at 250–300 ms after stimulus onset. This surprising finding might indicate an interaction between automatic semantic activation and inhibition of verbal responses required in no-go trials. Overall, the results suggest a close relationship between attentional processes and the manual motor system. Further research should clarify the functional role of the manual motor system activation in processing spatial information.

## Acknowledgments

I thank Martin H. Fischer for his valuable feedback on the early versions of this manuscript.

## Author Contributions

**Conceptualization:** A. Miklashevsky.

**Data curation:** A. Miklashevsky.

**Formal analysis:** A. Miklashevsky.

**Investigation:** A. Miklashevsky.

**Methodology:** A. Miklashevsky.

**Project administration:** A. Miklashevsky.

**Resources:** A. Miklashevsky.

**Software:** A. Miklashevsky.

**Validation:** A. Miklashevsky.

**Visualization:** A. Miklashevsky.

**Writing – original draft:** A. Miklashevsky.

**Writing – review & editing:** A. Miklashevsky.

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
