## [Decision Letter · Decision Letter 0]

17 Feb 2022

PONE-D-21-39874Catch the star! Spatial information activates the manual motor systemPLOS ONE

Dear Dr. Miklashevsky,

Thank you for submitting your manuscript to PLOS ONE. After careful consideration, we feel that it has merit but does not fully meet PLOS ONE’s publication criteria as it currently stands. Therefore, we invite you to submit a revised version of the manuscript that addresses the points raised during the review process. Please submit your revised manuscript by Apr 03 2022 11:59PM. If you will need more time than this to complete your revisions, please reply to this message or contact the journal office at plosone@plos.org. Please include the following items when submitting your revised manuscript:A rebuttal letter that responds to each point raised by the academic editor and reviewer(s). You should upload this letter as a separate file labeled 'Response to Reviewers'.A marked-up copy of your manuscript that highlights changes made to the original version. You should upload this as a separate file labeled 'Revised Manuscript with Track Changes'.An unmarked version of your revised paper without tracked changes. You should upload this as a separate file labeled 'Manuscript'.

We look forward to receiving your revised manuscript.

Kind regards,

Andriy Myachykov, PhD

Academic Editor

PLOS ONE

Journal Requirements:

Reviewers' comments:

Reviewer's Responses to Questions

**Comments to the Author**

1. Is the manuscript technically sound, and do the data support the conclusions?

Reviewer #1: Yes

Reviewer #2: Yes

2. Has the statistical analysis been performed appropriately and rigorously? 

Reviewer #1: Yes

Reviewer #2: Yes

3. Have the authors made all data underlying the findings in their manuscript fully available?

Reviewer #1: Yes

Reviewer #2: Yes

4. Is the manuscript presented in an intelligible fashion and written in standard English?

Reviewer #1: Yes

Reviewer #2: Yes

5. Review Comments to the Author

Reviewer #1: REVIEW – Catch the star! Spatial information activates the manual motor system

The study aims to evaluate the relationship between the manual motor system and exogenous and endogenous attention, the latter associated with two different forms of processing.

It is a very interesting manuscript, well written, addressing a relevant subject, and seeking evidence of important connections between motor function, central processing of sensory and motor information and attentional response.

I only have a few considerations about it, described below by section.

Introduction

The introduction was very well organized, presenting the theoretical framework in a structured and direct manner. The citations are current and relevant, properly building the line of reasoning that culminates in the presentation of the objective of the study. The reasons for investigating the relationship of the manual motor system with endogenous attention through two experiments, however, were not presented, causing a break in the development of the introduction. In this sense, lines 339-345 should be part of the introduction to justify the decision to carry out two experiments involving endogenous attention.

Methods and Results

The decision to present each experiment individually facilitates understanding. However, some aspects of the Methods that are valid for two or three experiments could be grouped together at the beginning of this section or presented in a single Table to allow a better visualization of the details of each experiment, e. g. exclusion of participants, exclusion of trials, accuracy, 5000 permutations performed in the cluster permutation analysis. There is no reason to describe the dips as they will not be discussed at any time. If you believe that these data are particularly important, Figures 2, 5, and 7 can present them satisfactorily, thus bringing more fluidity to the reading of the text.

The supplementary material brings the images of the stars and arrows as they were presented to the participants. However, there is no image for experiment 3. I was not able to open the words.osexp file, and the manuscript text does not provide information regarding the font, serif, and word style.

Furthermore, in experiments 1 and 2, the stimuli were presented in two colors, one of which was associated with the verbal task. However, in experiment 3, in addition to colors, there are two types of text (words and symbols). Thus, the data sample for each condition dropped from 90 in experiments 1 and 2 to 60 in experiment 3, the same experiment that had a 35% drop in sample size. What is the point of introducing symbols arrays in Experiment 3, in red and yellow, if these constructions did not contribute to the discussion and were not statistically analyzed?

In addition, the sample has participants with different levels of consistency in manuality. Did this feature have no impact on the results? Especially in relation to experiment 3, due to its smaller sample, what was the impact of the sample low consistency in manuality? What was the behavior of the left-handed portion of the sample regarding the results obtained? Wouldn't removing these participants from the sample imply more consistent results?

At various times, you classify your findings as close to significance. This is especially noteworthy in experiment 2, when a result close to significance reached p=.1. What is the threshold for considering a result close to or far from significance? Would considering these values as not significant significantly change your results? I believe that the text needs to be better worked on this point.

Although you present the conditional r and the marginal r of each calculation and experiment, you do not comment on the relative value of their different values throughout the study. Could you comment more on this?

Discussion

In the discussion, you say that the force increased for one hand and decreased for the other, as per the experiment. However, in the graphs, it is not possible to observe a drop in force in relation to the moments prior to the peak force zone, that is, there is not a valley and a peak occurring simultaneously when observing the tracing for each hand in Figures 2C and 5C. Figure 7C is quite confusing as to the line of each condition in the interaction between Hand and Word area.

Is there really a drop in the force exerted by the contralateral hand in experiments 1 and 2 and by the ipsilateral hand in experiment 3? Is it not possible that the increase in force is simply happening later, as appears to be represented in Figure 7C? Please explain more about this.

Similarly, the hypothesis regarding the grip force increasing of the contralateral hand in experiment 3 lacks further detail. The text seeks to explain that the inhibitory stimulus for the execution of the movement of the ipsilateral hand has a more pronounced effect in experiment 3 due to a greater semantic processing. However, what would lead to an increase of the grip force of the contralateral hand? Or is this increase just relative? I believe this is the most awaited part of the discussion, and, as it stands, it is not at the level of the work performed with so much labor and perfection described in this manuscript.

Figures

In the text, the caption of figure 7 states that the interaction area between Hand and Word occurs between 830-860 ms, but, from the figure, I believe it occurs between 230-260 ms.

The Figures 2C, 5C and 7C are very important to the manuscript and deserve a better treatment. The color code used for the interaction areas is confused with the colors adopted for the hands in different conditions. Likewise, the use of dashed and dotted lines is confused in the most important areas of these figures. Why not highlight the first 250 ms on a separate graph?

Finally, the colors adopted by experiment in figure 9 should not be the same colors adopted for each hand in the other figures.

Reviewer #2: The author presents the results of three experiments that measured the effects of lateralized displays of stimuli, symbolic stimuli indicating lateral regions of space, and semantic stimuli about different regions of lateralized space. They show that lateralized stars and central arrows reliably influence ipsilateral grip force with a characteristic timecourse, while words do so in the opposite way with a later timecourse. The studies are well designed and are informative and will be of especial interest in the use of the grip force sensory as a dependent measure. I think the results are clearly publishable and of interest. I especially appreciate the author’s calibration of their motivation and rationale with their interpretations, the lack of over interpretation, and the transparency regarding the exploratory nature of the study. I think there is much potential interest in the use of grip force as a behavioural measure of dynamic cognitive processing.

However, there are a couple of major and a few minor points where clarity would be beneficial, specifically regarding choices and procedures in the analysis pipeline and the presentation of the results. I have commented on these below. I think this is really an issue of clarifying for the reader some of the details that are missing in rationale and analytical strategy, and once sufficiently addressed, should be ready for publication. I present them in order of appearance below.

57: why is the sound presentation not reported?

94: what were participants told with respect to the sensors (what are they for) and how to hold them? And what were the instructions and participant’s told about the premise of the experiment (i.e. was a cover story used at all?).

106: how much calibration variation was there? i.e. is it pretty easy to calibrate? Please clarify.

141: the ‘global drift’ is drift within the practiced range (ie.. within the ‘grey’ range)?

“The global drift in force across the experiment was corrected by subtracting the average force from 20 ms intervals before stimulus onset from each epoch.:

Also, it is just a bit unclear to me mathematically what is happening in this sentence (i.e. what I do with the 20 ms intervals and how many there are and how they are applied to the epoch etc). Please clarify.

174: A bit more detail of how the permutations were done would be useful. I am assuming differences scores were used to test against 0? It seems clear how the main effects would be calculated, but were interactions calculated as differences of difference scores? More details about how the permutations test were conducted would be beneficial.

176: what kind of package is this?

181: The major point of clarification is this: What is the purpose of identifying significance with the permutation tests and then using LME? Why not use one or the other? I assume that the idea is that a visual inspection of the waveforms is not enough to find epochs of interest? But if permutation tests were used across the entire epoch, isn’t all the statistical inference you need in the results of your permutation tests? That is, isn’t using LME on already significant regions a bit of a redundant, double dip into the data? Perhaps more details about the permutation tests and its motivation, and the motivation for following up with LME will help clear this up. However, as of right now, it appears that the author could simply be using one or the other analytical strategy.

185: where is the drop1 function found?

192: please verify which effects size calculation was used (i.e. which package); also figure 2c could benefit from an in-figure legend.

207: please clarify why mean-center force of the opposite hand is used here and provide more details about the reasoning. Is this the DV that is used to generate fig 3 or is the ‘raw’ value used?

231: ‘see table N’ should be corrected

241: the summary here seems to capture the patterns in fig 3 but is a bit sparse. Doesn’t the figure show exactly what the author is seeking to show? Also, it is not clear to me how this plot was generated given the transformations to the data; please clarify.

257: there were two arrows presented centrally? Please clarify in text.

367: see table 1 for stimuli; this is a call to the wrong table. Also, why were meaningless symbol arrays used and how where they used? This is an added stimulus feature. Was it simply to lengthen the experiment? Please clarify.

527: the premise of the study is that grip force reflects these ‘cognitive factors’ and so this sentence is a bit odd here. The discussion starting at 538 seems to be the most relevant and so it might be beneficial to start the discussion with this part, and save the earlier speculations for later in the manuscript.

616: Note that here and throughout, the suggestion that this study is a study of the relationship between spatial attention and the manual motor system is undermined by some of the author’s speculations about the role of semantic/symbol processing (especially in the account of the effects observed with the word stimuli). Indeed, the author explains why their study is not like other studies using the Posner paradigm. It thus seems that the study is a study of the relationship between attention, motor system, and semantics/symbol processing.

6. PLOS authors have the option to publish the peer review history of their article (what does this mean?). If published, this will include your full peer review and any attached files.

Reviewer #1: No

Reviewer #2: **Yes: **Heath Matheson

---

## [Author Response · Author response to Decision Letter 0]

6 Apr 2022

Dear Reviewers,

Thank you for your valuable recommendations and suggestions. I added my replies in italic font after each point in the following text.

Reviewer #1: 

REVIEW – Catch the star! Spatial information activates the manual motor system

The study aims to evaluate the relationship between the manual motor system and exogenous and endogenous attention, the latter associated with two different forms of processing.

It is a very interesting manuscript, well written, addressing a relevant subject, and seeking evidence of important connections between motor function, central processing of sensory and motor information and attentional response.

I only have a few considerations about it, described below by section.

Introduction

The introduction was very well organized, presenting the theoretical framework in a structured and direct manner. The citations are current and relevant, properly building the line of reasoning that culminates in the presentation of the objective of the study. The reasons for investigating the relationship of the manual motor system with endogenous attention through two experiments, however, were not presented, causing a break in the development of the introduction. In this sense, lines 339-345 should be part of the introduction to justify the decision to carry out two experiments involving endogenous attention.

I shifted lines 339-345 to the introduction.

Methods and Results

The decision to present each experiment individually facilitates understanding. However, some aspects of the Methods that are valid for two or three experiments could be grouped together at the beginning of this section or presented in a single Table to allow a better visualization of the details of each experiment, e. g. exclusion of participants, exclusion of trials, accuracy, 5000 permutations performed in the cluster permutation analysis. There is no reason to describe the dips as they will not be discussed at any time. If you believe that these data are particularly important, Figures 2, 5, and 7 can present them satisfactorily, thus bringing more fluidity to the reading of the text.

Thank you for these suggestions. I already described most details relevant for all three experiments in the General Method section. Since exclusion of trials or the number of permutations require deeper explanations of the method (force thresholds and cluster permutation analysis), I decided to provide this information in smaller portions in a narrative manner. I believe this will facilitate understanding for readers who are not familiar with these methods.

Following your suggestion, I removed information about force dips from the text.

The supplementary material brings the images of the stars and arrows as they were presented to the participants. However, there is no image for experiment 3. I was not able to open the words.osexp file, and the manuscript text does not provide information regarding the font, serif, and word style.

Opening .osexp-files requires OpenSesame [https://osdoc.cogsci.nl] to be installed. I added this information to the supplementary file on OSF. I also added detailed information about font to the manuscript: “Participants saw words or stimuli symbol arrays having a length of around 2.5 cm (2.39 degrees of visual angle) in a 25 px (19 pt) Droid Sans Mono font.” (line 392)

Furthermore, in experiments 1 and 2, the stimuli were presented in two colors, one of which was associated with the verbal task. However, in experiment 3, in addition to colors, there are two types of text (words and symbols). Thus, the data sample for each condition dropped from 90 in experiments 1 and 2 to 60 in experiment 3, the same experiment that had a 35% drop in sample size. What is the point of introducing symbols arrays in Experiment 3, in red and yellow, if these constructions did not contribute to the discussion and were not statistically analyzed?

The original motivation for adding symbol arrays was to control for potential lexicality effects (cf. in EEG research: https://doi.org/10.1016/j.bandl.2008.12.001). I.e., it was possible that words lead to a specific pattern of grip force, and introducing non-word stimuli would then reveal word-specific force oscillations. However, it was not the case: all stimuli resulted in a similar force pattern. While I performed additional analysis for non-word stimuli, it did not help interpret the results related to the main hypothesis of the study. Thus, I decided not to include these distracting details in the manuscript.

Even though the data sample dropped to 60 trials per condition in Experiment 3, this number is still much larger than those used in previous force registration studies (e.g., https://doi.org/10.1371/journal.pone.0050287). Moreover, unlike linguistic studies with varying words and sentences, the present study repeated just three stimuli of interest (“left”, “center”, and “right”), which should additionally reduce variability of the dependent measure.

In addition, the sample has participants with different levels of consistency in manuality. Did this feature have no impact on the results? Especially in relation to experiment 3, due to its smaller sample, what was the impact of the sample low consistency in manuality? What was the behavior of the left-handed portion of the sample regarding the results obtained? Wouldn't removing these participants from the sample imply more consistent results?

I performed an exploratory analysis with and without a left-handed subsample, but there were no qualitative changes in the results. Due to the small size of the left-handed subsample (N = 4) it was not meaningful to analyze their data separately. 

At various times, you classify your findings as close to significance. This is especially noteworthy in experiment 2, when a result close to significance reached p=.1. What is the threshold for considering a result close to or far from significance? Would considering these values as not significant significantly change your results? I believe that the text needs to be better worked on this point.

For the mixed linear modeling method, I used a significance threshold of p = .1 to keep the variables in the model (cf. the recommendations in https://doi.org/10.1016/j.bandl.2021.104941), while always explicitly mentioning such effects as “close to significance” in the text. The only threshold qualifying effects to be significant was the standard of p = .05.

Considering “close to significance” values as insignificant would not substantially change the results of the study. The main effects of Position (Model 1.3), Direction (Models 2.1 and 2.2), or Hand (Model 3.1) that were close to significance are of less interest for the main hypothesis. Moreover, effects of Position and Direction have different signs for different hands, as further detailed analysis shows; averaging them thus necessarily reduces the overall level of significance. Most critical for the research hypothesis are effects found in each hand separately, such as in Model 1.4. Among those, there was only one effect with a p-value above the standard significance level (effect of star position in the left hand at 60-130 ms, Model 1.3; p = .052). In this case, however, the effect of Position was significant in the right hand in the same time window (p = .005). Also, it became significant in the left hand in the following time window of 260-100 ms (p = .011), with the same qualitative pattern. This converging evidence indicates that the effect was already present in the left hand at 60-130 ms and thus should be interpreted as real and not just a statistical artifact.

Although you present the conditional r and the marginal r of each calculation and experiment, you do not comment on the relative value of their different values throughout the study. Could you comment more on this?

I explained these values when describing Model 1.1: “Marginal r-squared (variance explained by fixed effects, see Nakagawa & Schielzeth, 2013) was .041, and conditional r-squared (variance explained by the whole model, i.e., fixed and random effects together) was .451.”

Discussion

In the discussion, you say that the force increased for one hand and decreased for the other, as per the experiment. However, in the graphs, it is not possible to observe a drop in force in relation to the moments prior to the peak force zone, that is, there is not a valley and a peak occurring simultaneously when observing the tracing for each hand in Figures 2C and 5C. Figure 7C is quite confusing as to the line of each condition in the interaction between Hand and Word area.

Is there really a drop in the force exerted by the contralateral hand in experiments 1 and 2 and by the ipsilateral hand in experiment 3? Is it not possible that the increase in force is simply happening later, as appears to be represented in Figure 7C? Please explain more about this.

Thank you for pointing out this invalid formulation. Indeed, while I referred to the results from linear modeling, it would be more appropriate to talk here about absolute values. The force always increases regardless of the condition, as the general pattern for each experiment demonstrates (Figures 2A, 5A, and 7A). I meant here (also corrected in the manuscript on page 29): 

“The force increased larger in each of the hands when the star was presented on the ipsilateral side; this increase was smaller for stars presented on the contralateral side.” 

The corresponding correction has also been made regarding the other two experiments (pp. 29-30).

Similarly, the hypothesis regarding the grip force increasing of the contralateral hand in experiment 3 lacks further detail. The text seeks to explain that the inhibitory stimulus for the execution of the movement of the ipsilateral hand has a more pronounced effect in experiment 3 due to a greater semantic processing. However, what would lead to an increase of the grip force of the contralateral hand? Or is this increase just relative? I believe this is the most awaited part of the discussion, and, as it stands, it is not at the level of the work performed with so much labor and perfection described in this manuscript.

Indeed, as I stated above, this was an unfortunate expression: there is no decrease but a relative increase in all cases. I interpreted this surprising finding as a conflict between word semantics and required vocal response: “The stronger H250 wave probably results from the conflict between the two overlapping processes: automatic activation of word semantics and conscious inhibition of verbal response. By suppressing semantics of the word “left”, participants literally had to inhibit activity in the left hand (for more details on the interplay between the semantic and motor system, see the HANDLE model, 54); the same happened with the word “right” and the right hand.”

Note that I added a reference to the work by García and Ibáñez (2016) where they discuss differing directions of the motor effects in embodied cognition research, depending on the timing, motor, and linguistic demands. Nevertheless, at this point, this is just a hypothetical post-hoc explanation since there are not enough studies using grip force registration in numerical cognition research. That is why I added the following suggestion to the General Discussion (p. 30):

“To further examine this hypothesis, it would be necessary to design a study following the procedure of Experiment 3 but with a non-linguistic response. I predict that words will exhibit force patterns similar to those in Experiments 1 and 2 in such a study since no inhibition of motor semantics should happen.”

Figures

In the text, the caption of figure 7 states that the interaction area between Hand and Word occurs between 830-860 ms, but, from the figure, I believe it occurs between 230-260 ms.

Thank you for pointing this out. It should be 250-300 ms. I corrected the caption.

The Figures 2C, 5C and 7C are very important to the manuscript and deserve a better treatment. The color code used for the interaction areas is confused with the colors adopted for the hands in different conditions. Likewise, the use of dashed and dotted lines is confused in the most important areas of these figures. Why not highlight the first 250 ms on a separate graph?

Finally, the colors adopted by experiment in figure 9 should not be the same colors adopted for each hand in the other figures.

Thank you for these suggestions. I changed the colors in figures and added three new bar charts (Figures 3, 7, and 10) representing grip force averaged by condition in the time windows of interest. As you suggested, I also changed the color coding in Fig. 9 (Fig. 12 in the new version of the manuscript).

Reviewer #2: 

The author presents the results of three experiments that measured the effects of lateralized displays of stimuli, symbolic stimuli indicating lateral regions of space, and semantic stimuli about different regions of lateralized space. They show that lateralized stars and central arrows reliably influence ipsilateral grip force with a characteristic timecourse, while words do so in the opposite way with a later timecourse. The studies are well designed and are informative and will be of especial interest in the use of the grip force sensory as a dependent measure. I think the results are clearly publishable and of interest. I especially appreciate the author’s calibration of their motivation and rationale with their interpretations, the lack of over interpretation, and the transparency regarding the exploratory nature of the study. I think there is much potential interest in the use of grip force as a behavioural measure of dynamic cognitive processing.

However, there are a couple of major and a few minor points where clarity would be beneficial, specifically regarding choices and procedures in the analysis pipeline and the presentation of the results. I have commented on these below. I think this is really an issue of clarifying for the reader some of the details that are missing in rationale and analytical strategy, and once sufficiently addressed, should be ready for publication. I present them in order of appearance below.

57: why is the sound presentation not reported?

Initially, I expected to find a similar effect of attention manipulation on grip force regardless of the stimulus type, perhaps with variability in the timing and magnitude of the effect as a function of stimulus. However, the force pattern generated by sounds was strikingly different from those resulting from other types of stimuli: sounds lead to a clear initial dip in force followed by one short and one very long peak, probably correlating in magnitude with sound intensity (see the image below). It seems that qualitatively different processes are reflected in grip force in this case. I decided to focus on visual stimuli only in order to keep the narrative and interpretation of results as clear as possible. Still, for transparency reasons, I reported in the manuscript that there was one more experiment in the study. I have added a brief explanation for why this work is not part of the present report (see Footnote 1 on p. 5 of the revised manuscript).

94: what were participants told with respect to the sensors (what are they for) and how to hold them? And what were the instructions and participant’s told about the premise of the experiment (i.e. was a cover story used at all?).

Figure 1C shows how participants held the sensors (the experimenter demonstrated this but did not strictly control during the study). 

As I described in the General Method section (lines 110-118), participants knew that sensors register their grip force. Importantly, as the text states, participants were explicitly instructed to keep the same force during the experiments, i.e., any force changes should not be related to the instruction. Moreover, the magnitude of force changes of interest is so small (in the range of 5-10 mN) that it is unlikely to result from participants’ conscious control. 

I also added the following statement to the General Method section (line 119):

“There was no cover story used for the participants.”

106: how much calibration variation was there? i.e. is it pretty easy to calibrate? Please clarify.

I added the following text (lines 117-118):

“Most participants successfully learned to perform the calibration within 15-30 seconds. There were, however, a few participants who required up to two minutes during the first calibration.”

141: the ‘global drift’ is drift within the practiced range (ie.. within the ‘grey’ range)?

“The global drift in force across the experiment was corrected by subtracting the average force from 20 ms intervals before stimulus onset from each epoch.:

Also, it is just a bit unclear to me mathematically what is happening in this sentence (i.e. what I do with the 20 ms intervals and how many there are and how they are applied to the epoch etc). Please clarify.

I reformulated this passage as follows: 

“The global variation in force across the experiment was corrected by (1) averaging force within a 20 ms interval before stimulus onset in each epoch and (2) subtracting this average force from the entire epoch.”

I replaced the less clear term “drift” with a more straightforward “variation” and added Footnote 2:

“Under global force variation, I understand here stimulus-unrelated changes in force over longer periods of time, e.g., if participants press sensors stronger due to higher arousal at the beginning of the experiment or slightly release the sensors in the second half of the session due to fatigue, etc.”

174: A bit more detail of how the permutations were done would be useful. I am assuming differences scores were used to test against 0? It seems clear how the main effects would be calculated, but were interactions calculated as differences of difference scores? More details about how the permutations test were conducted would be beneficial.

I added Footnote 3:

“Cluster permutation analysis performs bootstrapping of continuous signals. In this analysis, conditions are randomly assigned to epochs, which results in a random data structure. According to newly assigned labels, a t-statistics is calculated. The mass of the clusters exceeding a significance threshold is stored. The procedure is repeated multiple times, with the resulting distribution of random cluster masses that can be found in the dataset. The actual cluster mass is then compared with bootstrapped cluster masses, and the likelihood of the observed result is calculated.

I suggest using this method in case of force registration for exploratory purposes to identify potential time windows of interest. Linear mixed-effects models are used in the present study for confirmatory analyses.”

For the current analysis, the following formula was used (see Supplementary materials at OSF):

clusterlm (cluster_perm_data ~ 

hand*position + 

Error (participant/(hand*position)), 

data = cluster_perm_design, 

np = 5000, 

multcomp = c("tfce"))

176: what kind of package is this?

permuco is an R package designed for cluster permutation analysis of continuous data (such as EEG signal): https://cran.r-project.org/web/packages/permuco/index.html

181: The major point of clarification is this: What is the purpose of identifying significance with the permutation tests and then using LME? Why not use one or the other? I assume that the idea is that a visual inspection of the waveforms is not enough to find epochs of interest? But if permutation tests were used across the entire epoch, isn’t all the statistical inference you need in the results of your permutation tests? That is, isn’t using LME on already significant regions a bit of a redundant, double dip into the data? Perhaps more details about the permutation tests and its motivation, and the motivation for following up with LME will help clear this up. However, as of right now, it appears that the author could simply be using one or the other analytical strategy.

Previous studies using force registration either analyzed predefined time windows based on a priori hypotheses (e.g., https://journals.plos.org/plosone/article?id=10.1371/journal.pone.0050287) or cut the epoch into multiple equal time windows and analyzed each of them separately (e.g., https://www.ijert.org/research/portable-device-validation-to-study-the-relation-between-motor-activity-and-language-verify-the-embodiment-theory-through-grip-force-modulation-IJERTV5IS120003.pdf).

The advantage of cluster permutation analysis is that it identifies significant differences between conditions even without any prior knowledge and simultaneously accounts for the multiple comparison problem. It makes this technique very useful for exploration of continuous data. However, in my view, this analysis is too conservative in the case of noisy force data if used for confirmatory purposes.

On the other hand, simply cutting the epoch into multiple time windows, even if accounting for multiple comparisons, is not optimal: (1) it does not give enough precision with regard to the timing of effects, and (2) it might mask significant effects which start at the end of one epoch and end at the beginning of the next epoch since they become averaged with portions of irrelevant data. 

That is why I suggest using the cluster permutation analysis as a purely exploratory technique helping to identify the time windows with potential effects. Yet, more sophisticated methods (mixed models using covariates, e.g., see Models 1.3 and 1.4) are the only basis for my claims about the presence or absence of significant findings in the current study.

185: where is the drop1 function found?

The drop1 function is part of the lme4 package in R, a standard tool for linear mixed-effect modelling.

192: please verify which effects size calculation was used (i.e. which package); also figure 2c could benefit from an in-figure legend.

I did not calculate the effect size a priori, and I am not aware of power calculation tools for mixed models. Nevertheless, Brysbaert and Stevens (2018; https://www.journalofcognition.org/articles/10.5334/joc.10/) recommend 1,600 observations per condition for standard RT studies, and they argue that mixed-effect analyses are more powerful than usual separate analyses by participants and by items. There were 3,690 observations in my Experiments 1 and 2, and 1,620 observations in Experiment 3. In that way, I tried to follow the recommendations by Brysbaert and Stevens (2018).

While I agree that Figures 2C, 6C, and 9C (in the new version) would benefit from in-figure legends, I decided to magnify the force patterns and describe all six conditions in figure captions instead in order to avoid overloading the images with visual information.

207: please clarify why mean-center force of the opposite hand is used here and provide more details about the reasoning. Is this the DV that is used to generate fig 3 or is the ‘raw’ value used?

I extended Footnote 4 as follows:

“The force of the opposite hand was used as a predictor in order to account for the correlation of forces due to automatic coordination between hands (Mathew et al., 2020). I expect lateralized stimuli to lead to increased force on the ipsilateral side. Still, at the same time, it is clear from force patterns that the force of both hands changes simultaneously, i.e., when the force of one hand increases, so does the other (see Figures 2C, 6C, and 9C). It implies that this basic physiological mechanism might mask lateralized effects of interest. That is why I suggest using the contralateral force as a covariate and estimating the effect of interest beyond the variance explained by the contralateral force.”

To make intercepts in regressions meaningful, I centered all continuous predictors prior to using linear modeling. In each analysis, dependent variables are not mean-centered. Thus, Figures 4, 5, 8, and 11 are based on “raw” values calculated with mean-centered predictors/covariates. 

231: ‘see table N’ should be corrected

Thank you for pointing it out. I corrected this mistake.

241: the summary here seems to capture the patterns in fig 3 but is a bit sparse. Doesn’t the figure show exactly what the author is seeking to show? Also, it is not clear to me how this plot was generated given the transformations to the data; please clarify.

As I explained above, the regression lines are extracted from linear models. Of interest here, I added figures with raw force values by condition, as requested by Reviewer 1 (Figures 3, 7, and 10 in the new version). I hope this makes the situation clear for readers. As you suggested, I extended the summary as follows:

“To summarize, Experiment 1 demonstrated an early (already at 60-130 ms after stimulus onset) interaction between Hand and Position: lateralized stimuli led to relatively stronger force on the ipsilateral side. However, the effect of lateralized stimuli on grip force was rather asymmetric: the effect of Position was more pronounced in the right hand than in the left hand. The same pattern emerged at 260-1000 ms, with even higher significance in both hands.”

257: there were two arrows presented centrally? Please clarify in text.

Participants always saw only one arrow at a time. I resolved this ambiguity in the text. Thank you.

367: see table 1 for stimuli; this is a call to the wrong table. Also, why were meaningless symbol arrays used and how where they used? This is an added stimulus feature. Was it simply to lengthen the experiment? Please clarify.

Thank you, I removed the reference to the (non-existing) table. 

The original motivation for adding symbol arrays was to control for potential lexicality effects (cf. in EEG research: https://doi.org/10.1016/j.bandl.2008.12.001). I.e., it was possible that words lead to a specific pattern of grip force, and introducing non-word stimuli would then reveal word-specific force oscillations. However, it was not the case: all stimuli resulted in a similar force pattern. While I performed additional analysis for non-word stimuli, it did not help interpret the results related to the study's main hypothesis. Thus I decided not to include these distracting details in the manuscript.

527: the premise of the study is that grip force reflects these ‘cognitive factors’ and so this sentence is a bit odd here. The discussion starting at 538 seems to be the most relevant and so it might be beneficial to start the discussion with this part, and save the earlier speculations for later in the manuscript.

Most previous studies using force registration assumed that grip force changes are caused exclusively by motor semantics of stimuli or action observation (motor resonance). One crucial finding in my studies using this technique is that grip force demonstrates systematic oscillations regardless of stimulus type, i.e., this holds also for motor-unrelated stimuli. I believe this finding and its methodological implications deserve a more detailed discussion which is outside of the scope of the current paper. Yet, it is essential to emphasize that force data might reflect more general processes (such as stimulus identification or response inhibition). While I recognize that these considerations are rather speculative and require further empirical support, I would prefer to keep them at the beginning of the discussion as they help interpret the timing of primary effects of interest.

616: Note that here and throughout, the suggestion that this study is a study of the relationship between spatial attention and the manual motor system is undermined by some of the author’s speculations about the role of semantic/symbol processing (especially in the account of the effects observed with the word stimuli). Indeed, the author explains why their study is not like other studies using the Posner paradigm. It thus seems that the study is a study of the relationship between attention, motor system, and semantics/symbol processing.

It is correct with respect to arrows and words: while these are symbolic stimuli, the relationship between the visual input and observed force patterns must be mediated by semantic/symbolic processing. However, presentation of stars (Experiment 1) necessarily led to attentional shifts per se. It is exactly what makes all three experiments a single study: I expected all three kinds of stimuli to cause the same effect by different means. Moreover, the timing of the interaction between side and hand followed my initial expectations: this interaction emerged very early for physical stimuli (Experiment 1) but later for symbolic stimuli (Experiments 2 and 3), which require more time for extraction of meaning. I added this clarification to the end of Introduction (lines 60-63).

---

## [Decision Letter · Decision Letter 1]

4 May 2022

PONE-D-21-39874R1Catch the star! Spatial information activates the manual motor systemPLOS ONE

Dear Dr. Miklashevsky,

Thank you for submitting your manuscript to PLOS ONE. After careful consideration, we feel that it has merit but does not fully meet PLOS ONE’s publication criteria as it currently stands. Therefore, we invite you to submit a revised version of the manuscript that addresses the points raised during the review process.

We look forward to receiving your revised manuscript.

Kind regards,

Andriy Myachykov, PhD

Academic Editor

PLOS ONE

Journal Requirements:

Additional Editor Comments (if provided):

Dear authors,

Thank you for submitting the revised version of your manuscript titled "Catch the star! Spatial information activates the manual motor system". I am sure you will be happy to see that Reviewer 2 finalized their review and that Reviewer 1 only has one outstanding issue with the revised submission. In general, I agree with the Reviewer that a form of discloure is necessary regarding the the no-word stimuli results.

Kind regards,

A. Myachykov

Reviewers' comments:

Reviewer's Responses to Questions

**Comments to the Author**

1. If the authors have adequately addressed your comments raised in a previous round of review and you feel that this manuscript is now acceptable for publication, you may indicate that here to bypass the “Comments to the Author” section, enter your conflict of interest statement in the “Confidential to Editor” section, and submit your "Accept" recommendation.

Reviewer #1: All comments have been addressed

Reviewer #2: All comments have been addressed

2. Is the manuscript technically sound, and do the data support the conclusions?

Reviewer #1: Yes

Reviewer #2: Yes

3. Has the statistical analysis been performed appropriately and rigorously? 

Reviewer #1: Yes

Reviewer #2: Yes

4. Have the authors made all data underlying the findings in their manuscript fully available?

Reviewer #1: Yes

Reviewer #2: Yes

5. Is the manuscript presented in an intelligible fashion and written in standard English?

Reviewer #1: Yes

Reviewer #2: Yes

6. Review Comments to the Author

Reviewer #1: The changes made to the article were punctual and well-directed, so I believe the article has evolved substantially. There is still, in my opinion, one point to be resolved, but I leave it to the Editor to judge whether my advice is adequate or not. I have placed my comments in dark blue in the attached document.

Reviewer #2: (No Response)

7. PLOS authors have the option to publish the peer review history of their article (what does this mean?). If published, this will include your full peer review and any attached files.

Reviewer #1: No

Reviewer #2: **Yes: **Heath Matheson

---

## [Author Response · Author response to Decision Letter 1]

11 May 2022

Reviewer #1: 

R1 – Although you have decided not to include the no-word stimuli results in the manuscript, you have decided to include its description. There is no problem about not having a hypothesis validated by the results, but it is not interesting that the data are not presented because they do not agree with that given hypothesis. I suggest exposing such data and discussing this topic in the manuscript - the hypothesis, the results, and their implication in the study - or removing this detail from the description of the experiment.

REPLY: Thank you for this recommendation. I updated the manuscript accordingly: 

Lines 376-378: “Symbol arrays were included to investigate potential lexicality effects, i.e., the difference in signal between word and non-word stimuli found, for example, in EEG at 150-250 ms after stimulus onset (Pulvermüller et al., 2009).”

Fig. 9A: I plotted a separate line for no-go trials with symbol arrays.

Lines 402-404: “Note that the pattern for symbol arrays closely resembles that for words, especially in the critical time window around 200 ms. The only difference between the two forces is the larger peak for symbol arrays around 650 ms.”

Lines 418-421: “As before, a cluster permutation analysis was used for exploratory purposes. First, Hand (right / left) and Lexicality (word / symbol array) as within-variables and their interaction were submitted. The analysis did not reveal any significant effects (all p-values > .72). This result indicates that lexicality is not reflected in the force signal; trials with symbol arrays were excluded from all further analyses.”

Lines 619-624 (Discussion): “An additional finding in Experiment 3 is that lexicality, i.e., the distinction between word and non-word stimuli, previously found to influence the EEG signal at 150-250 ms after stimulus onset (Pulvermüller et al., 2009), is not reflected in grip force signal. Unlike in most previous studies on lexical processing, meaningless symbol arrays (§@#$%) were used in the present study instead of pseudowords. Future research should consider using pseudowords built according to phonetic and morphological regularities of the language under investigation to confirm the present finding.”

I also updated the OSF materials by adding two scripts for interested readers:

• Experiment 3: Words / 04 processing scripts / 011_cluster permutation_prepare_lexicality.R

• Experiment 3: Words / 04 processing scripts / 012_cluster_permutation_analysis_lexicality.R

---

## [Editor Report · Decision Letter 2]

18 May 2022

Catch the star! Spatial information activates the manual motor system

PONE-D-21-39874R2

Dear Dr. Miklashevsky,

We’re pleased to inform you that your manuscript has been judged scientifically suitable for publication and will be formally accepted for publication once it meets all outstanding technical requirements.

Kind regards,

Andriy Myachykov, PhD

Academic Editor

PLOS ONE
---

## [Editor Report · Acceptance letter]

29 Jun 2022

PONE-D-21-39874R2 

Catch the star! Spatial information activates the manual motor system 

Dear Dr. Miklashevsky:

I'm pleased to inform you that your manuscript has been deemed suitable for publication in PLOS ONE. Congratulations! Your manuscript is now with our production department. 

Kind regards, 

on behalf of

Dr. Andriy Myachykov 

Academic Editor

PLOS ONE